# Radiocontrast Agent Diatrizoic Acid Induces Mitophagy and Oxidative Stress via Calcium Dysregulation

**DOI:** 10.3390/ijms20174074

**Published:** 2019-08-21

**Authors:** Dakota B. Ward, Kathleen C. Brown, Monica A. Valentovic

**Affiliations:** Department of Biomedical Sciences, Toxicology Research Cluster, Joan C. Edwards School of Medicine, Marshall University, 1 Marshall Drive, Huntington, WV 25755, USA

**Keywords:** contrast-induced acute kidney injury, diatrizoic acid, proximal tubule cytotoxicity, Seahorse XFe, HK-2 cells, mitophagy, oxidative stress

## Abstract

Contrast-induced acute kidney injury (CI-AKI) is the third most common cause of hospital associated kidney damage. Potential mechanisms of CI-AKI may involve diminished renal hemodynamics, inflammatory responses, and direct cytotoxicity. The hypothesis for this study is that diatrizoic acid (DA) induces direct cytotoxicity to human proximal tubule (HK-2) cells via calcium dysregulation, mitochondrial dysfunction, and oxidative stress. HK-2 cells were exposed to 0–30 mg I/mL DA or vehicle for 2–24 h. Conversion of 3-(4,5-dimethylthiazol-2-yl)-2,5-diphenyltetrazolium bromide (MTT) and trypan blue exclusion indicated a decrease in mitochondrial and cell viability within 2 and 24 h, respectively. Mitochondrial dysfunction was apparent within 8 h post exposure to 15 mg I/mL DA as shown by Seahorse XF cell mito and Glycolysis Stress tests. Mitophagy was increased at 8 h by 15 mg I/mL DA as confirmed by elevated LC3BII/I expression ratio. HK-2 cells pretreated with calcium level modulators BAPTA-AM, EGTA, or 2-aminophenyl borinate abrogated DA-induced mitochondrial damage. DA increased oxidative stress biomarkers of protein carbonylation and 4-hydroxynonenol (4HNE) adduct formation. Caspase 3 and 12 activation was induced by DA compared to vehicle at 24 h. These studies indicate that clinically relevant concentrations of DA impair HK-2 cells by dysregulating calcium, inducing mitochondrial turnover and oxidative stress, and activating apoptosis.

## 1. Introduction

The use of iodinated radiocontrast media (RCM) to visualize internal structures during diagnostic procedures has increased exponentially since their first use in 1928. RCM may lead to contrast-induced acute kidney injury (CI-AKI) which occurs in up to 30% of patients and is the third leading cause of iatrogenic acute renal failure accounting for 10–25% of all AKI cases [1]. Following X-ray based imaging procedures such as percutaneous coronary intervention and cardiac angiography, renal dysfunction can range from non-symptomatic increases in serum creatinine (SCr) to severe and permanent renal damage resulting in the need for dialysis [2]. The renal pathogenesis of CI-AKI is complex and not completely understood. Intravenous administration of RCM induces transient vasodilation of renal vasculature followed by severe, sustained vasoconstriction resulting in decreased oxygen supply and oxidative damage to the outer renal medulla [3,4]. Aside from the renal ischemia, exposure to RCM has been shown to be directly toxic to renal parenchyma [5,6,7]; however, the exact mechanisms of direct cytotoxicity have not been fully elucidated.

Numerous studies using very high concentrations of RCM have evaluated RCM cytotoxicity to the proximal tubule (PT). Concentrations in excess of clinically relevant RCM levels have characterized PT damage in vitro as vacuolization [8,9], increased production of reactive oxygen species (ROS), induction of oxidative stress, mitochondrial dysfunction, diminished ATP production, activation of the unfolded protein response (UPR) and endoplasmic reticulum (ER) stress, as well as alterations in stress kinases [8,9,10,11,12]. Although there has been substantial knowledge obtained from these studies, the initial onslaught of cellular damage remains elusive. This may be, in part, due to the clear majority of mechanistic studies using concentrations of RCM far higher than what PT cells experience in the healthy kidney. It is common place to use 100–200 mg I/mL RCM in vitro, equating to approximately 5–100 times the concentrations found in circulating plasma [13,14]. It is possible that exposing cells to concentrations of RCM of this magnitude are inducing changes in cellular homeostasis that would not occur at clinically relevant concentrations of RCM.

The purpose of this study was to explore the various mechanisms of cytotoxicity associated with the clinically relevant concentrations of the first-generation RCM diatrizoic acid (DA) in a human kidney (HK-2) cell line. This model was chosen as these are adult, non-cancerous, immortalized human epithelial cells that maintain biochemical properties and activities similar to in vivo proximal tubule cells [15,16,17]. Additionally, HK-2 cells demonstrate the organic ion transporters (OAT) 1 and 3, which have been reported to transport DA into proximal tubule cells [18]. The present study was undertaken to demonstrate that exposure of HK-2 cells to clinically relevant concentrations of DA induces cytotoxicity and affects various cellular mechanisms such as mitochondrial function, cytokine release and downstream activity, activates the UPR and ER stress, induces oxidative stress, and determines the role of calcium homeostasis in direct PT cytotoxicity.

## 2. Results

### 2.1. DA Effects on Mitochondrial and Cell Viability

Conversion of 3-(4,5-dimethylthiazol-2-yl)-2,5-diphenyltetrazolium bromide (MTT), to formazan was used as an indicator of mitochondrial viability. DA exposure induced concentration and time dependent changes in MTT (Figure 1). DA substantially reduced mitochondrial viability within 2 h (*p* < 0.001) beginning with the 15 mg I/mL DA when compared to vehicle control. MTT values were diminished at all DA concentrations at 8 h and 24 h (*p* < 0.001) when compared to vehicle control (Figure 1). A concentration-dependent decrease in mitochondrial viability was evident at 8 h and 24 h when compared to other treatment groups (*p* < 0.05) (Figure 1). A time-dependent decrease in mitochondrial viability was also evident when comparing the different exposure time points (*p* < 0.01) (Figure 1). Trypan blue exclusion was used as an indicator of cell viability and loss of membrane integrity, as well as confirmation that the DA mediated decline in MTT reduction was not due to a decrease in the overall number of viable cells. Unlike the MTT assay, there was no significant decrease in cell viability until 24 h exposure to concentrations of 23 mg I/mL DA or higher (*p* < 0.05) (Figure 2). DA final concentrations of 28 and 30 mg I/mL showed an additional decline in cell viability at 24 h when compared to other treatment groups (*p* < 0.05) (Figure 2). A time-dependent decrease in cell viability was also evident when compared to other time points (*p* < 0.01) (Figure 2). Thus, DA at clinically relevant concentrations, first decreased the conversion of MTT to formazan within 2 h and at 24 h caused loss of cell membrane integrity as indicated by trypan blue exclusion. These studies suggested that our model was appropriate to explore the cellular mechanisms of DA-induced cytotoxicity in HK-2 cells.

### 2.2. DA Effects on Mitochondrial Function and Energy Utilization

Mitochondrial function following exposure to DA was assessed using an Agilent Seahorse XFe instrument. In an attempt to more accurately understand the effects of DA on mitochondrial function, various XFe assays were utilized including: cell mito stress test, cell glycolysis stress test, mito fuel flex test, and real-time ATP rate assay. In the cell mitochondrial stress test, oxygen consumption rate (OCR) following serial injection of various probes was used as an indicator of mitochondrial function. Oligomycin, an ATP synthase inhibitor, probes for ATP linked oxygen consumption; carbonyl cyanide-4-(trifluoromethoxy)phenylhydrazone (FCCP), an oxidative phosphorylation uncoupling agent, induced maximum oxygen consumption and the resultant OCR was used to calculate spare respiratory capacity; and the final injection, a mixture of rotenone and antimycin-A inhibited complex I and complex III resulting in complete inhibition of mitochondrial respiration and determination of the non-mitochondrial oxygen consumption.

DA decreased basal OCR, maximal OCR, spare respiratory capacity, and ATP production at 8 and 24 h (Figure 3). OCR was decreased at 18 mg I/mL for basal OCR (*p* < 0.05), 15 mg I/mL for maximal OCR and spare respiratory capacity (*p* < 0.01), and 23 mg I/mL for ATP production (*p* < 0.05) when compared to vehicle control at 8 h; however, OCR linked to proton leak and non-mitochondrial oxygen consumption did not change (Figure 3). Basal OCR, maximal OCR, spare respiratory capacity, and ATP production were all significantly decreased at 15 mg I/mL (*p* < 0.001) within 24 h when compared to vehicle control (Figure 3). Similar to the 8 h timepoint, there was no change in non-mitochondrial oxygen consumption, although, there was an increase in OCR linked to proton leak at 30 mg I/mL (*p* < 0.05) when compared to vehicle control at 24 h (Figure 3) indicating damage to the mitochondrial inner membrane.

The cell glycolysis stress test utilizes extracellular acidification rate (ECAR) as an indicator of various parameters of glycolysis. HK-2 cells were starved of glucose and pyruvate for 40 min prior to glucose saturation and measurement of basal glycolysis. The ATP synthase inhibitor oligomycin was then injected to drive glycolysis to maximum capacity from which glycolytic capacity and glycolytic reserve could be calculated. ECAR linked to non-glycolytic acidification was determined by the addition of the hexokinase inhibitor, 2-deoxyglucose. Whereas decreases in OCR were evident at 15 mg I/mL at both 8 and 24 h, statistically significant decreases in glycolysis and glycolytic capacity as shown by ECAR were not apparent until 28 mg I/mL at 8 h (*p* < 0.05) and 23 mg I/mL at 24 h (*p* < 0.05) (Figure 4). Glycolytic reserve and non-glycolytic acidification were unchanged for all concentrations at both time points (Figure 4).

The initial studies examining mitochondrial function indicated DA impaired mitochondrial function prior to alterations in cell viability (Figure 2). Additional studies were conducted to probe alterations in mitochondrial function and energy production. The real-time ATP rate assay was used to evaluate DA changes in mitochondrial and glycolytic pathway production of ATP. The real-time ATP assays utilize serial injections of oligomycin and rotenone/antimycin-A to determine ATP production within the cell. OCR, ECAR, and proton efflux rate (PER) measurements were used to calculate total ATP production, glycolytic ATP production, and mitochondrial ATP production. Exposure to DA for 24 h induced a significant decrease in total and mitochondrial ATP production at 18 mg I/mL (*p* < 0.05) but the slight decrease in glycolytic ATP production was not significant (Figure 5). The findings suggest that the decline in total ATP production, therefore, can be attributed to the decrease in mitochondrial ATP production.

The mito fuel flex test measures OCR following inhibition of the three major mitochondrial fuel sources (glucose, glutamine, and fatty acid oxidation) to determine the mitochondrial dependency and flexibility on each fuel source. The mitochondrial dependency on a specific fuel source is determined by first injecting an inhibitor of the pathway in question followed by inhibition of the other two pathways. The flexibility of the mitochondria to adjust to changes in fuel sources is calculated by subtracting the dependency of a fuel source from the mitochondrial capacity of that fuel source. Capacity is determined by first inhibiting the other fuel pathways followed by the pathway in question. In response to exposure to DA for 8 h, there was no significant change in the dependency of glutamine, glucose, or fatty acid oxidation (Figure 6). The flexibility to glutamine and glucose oxidation was unchanged as well (Figure 6). These results suggest that DA does not inflict damage to the fuel transport or oxidation machinery within the mitochondria. 

### 2.3. DA Effects on Mitophagy 

A decrease in mitochondrial function in the absence of damage to mitochondrial fuel oxidation may be partially due to a decrease in the total number of active mitochondria. Consequently, studies were conducted to evaluate the role of DA exposure on the expression of microtubule-associated proteins 1A/1B light chain 3B I and II (LC3BI and II). LC3BI was decreased by DA relative to control within 8 h at 18–30 mg I/mL (*p* < 0.05) and to 15–30 mg I/mL within 24 h (*p* < 0.05) (Figure 7 and Figure 8). LC3BII expression showed a trend to increase within 8 h of exposure to DA but was not statistically different (Figure 7). The increase in LC3BII expression did increase to significance within 24 h (*p* < 0.01) (Figure 8). However, the relative ratio of LC3BII/I was significantly increased (*p* < 0.05, *p* < 0.01) at both time points (Figure 7 and Figure 8). These findings suggest that the decrease in OCR observed in the cell mito stress could be due to mitophagy and not damage to the electron transport chain (ETC) or transport machinery. This conclusion was further supported by the results of the mito fuel flex test.

### 2.4. DA Effects on Endoplasmic Reticulum Stress

Potential mechanisms for DA cytotoxicity to the PT may be mediated by alterations in cell repair and protein folding. To evaluate the role of DA exposure and the induction of the unfolded protein response (UPR) and ER stress, protein expression was measured for glucose-regulated protein 78 (GRP78) and CCAAT/enhancer-binding protein homologous protein (CHOP). DA had no effect on protein expression of GRP78 (Figure 9) or CHOP (Figure 10) expression at any time point. These results suggest that UPR and ER stress do not play a role in DA-induced cytotoxicity.

### 2.5. DA Effects on Oxidative Stress and Superoxide Dismutase (SOD)

DA induction of oxidative stress was measured by evaluating protein carbonylation and 4-hydroxynonenal (4HNE) adduct formation via Western blot. DA increased oxidative stress as shown by Oxyblot analysis at 24 h relative to vehicle control in groups treated with 18 mg I/mL DA (*p* < 0.05) (Figure 11). Protein carbonylation was not increased relative to control at 8 h or at lower concentrations at 24 h (Figure 11). DA also increased 4HNE adduct formation at 24 h relative to vehicle control in groups treated with 18–30 mg I/mL DA (*p* < 0.05) (Figure 12). Protein adduct formation of 4HNE was not increased relative to control at 8 h or at concentrations of 0–15 mg I/mL lower DA at 24 h (Figure 12). Exposure of HK-2 cells to clinically relevant concentrations of DA induced increases in 4HNE and protein carbonylation at 24 h but not 8 h, indicating that oxidative stress occurs as a result of previous cellular damage. 

MnSOD expression was measured to determine the effect of DA exposure on antioxidant systems within the mitochondria. MnSOD expression was comparable between control and DA groups at 24 h (Figure 13). Protein expression levels do not necessarily indicate activity of an enzyme; consequently, MnSOD, Cu/ZnSOD, and total SOD activity were measured to evaluate the effects of DA on these important cellular antioxidant enzymes. Total SOD activity was decreased at 24 h relative to vehicle control in groups treated with 28 and 30 mg I/mL (*p* < 0.05) (Figure 13). MnSOD activity was unchanged in response to exposure to DA at 24 h (Figure 13), while Cu/ZnSOD activity was significantly reduced in response to 23 mg I/mL DA (*p* < 0.05) relative to vehicle control at 24 h exposure (Figure 13). Protein carbonylation and 4HNE adduction were measured in cytosolic and mitochondrial fractions to determine if MnSOD activity within the mitochondria is protecting HK-2 mitochondria from DA-induced ROS generation. Following exposure to 30 mg I/mL DA for 24 h, protein carbonylation and 4HNE protein adduction was significantly increased in response to control within the cytosolic fraction but there was no change in either biomarker of oxidative stress in the mitochondrial fraction (Figure 14). 

To evaluate potential sources for ROS production, tumor necrosis factor alpha (TNFα) expression was measured in cell lysate and cell media via Western blot and ELISA, respectively. TNFα expression in cell lysate decreased at 24 h relative to vehicle control following exposure to 28 and 30 mg I/mL DA (*p* < 0.01) as shown by Western blot (Figure 15). TNFα secretion into the media was increased at 24 h above vehicle control levels (*p* < 0.05) by 30 mg I/mL (Figure 15). NADPH oxidase 4 (NOX4), a ROS generating enzyme linked to TNF-α, expression was also measured. NOX4 expression was unchanged in response to 24 h exposure to DA (Figure 15). Although it is apparent that DA exposure elicits an inflammatory response in HK-2 cells, activation of TNFα does not play a role in DA-induced oxidative stress at clinically relevant concentrations. Therefore, the source of ROS overproduction in response to DA remains elusive. 

### 2.6. DA Effects on Mitochondrial Membrane Leakage and Apoptosis Initiation 

Alterations in mitochondrial membrane integrity were evident following exposure to DA for 24 h (Figure 16). DA caused significant cytochrome c leakage from the mitochondrial inner membrane space into the cytosol relative to vehicle control (*p* < 0.05). Further, cytochrome c expression within mitochondrial fractions decreased at 30 mg I/mL relative to control (*p* < 0.05) as shown by Western blot (Figure 16). 

DA increased caspase 3 cleavage and caspase 12 expression relative to vehicle control at 28 and 30 mg I/mL following 24 h exposure (*p* < 0.05, *p* < 0.01) (Figure 17). Caspase 4 expression and cleavage was unchanged in response to exposure to DA for 24 h suggesting that the efflux of TNFα into the media was not sufficient to induce activation of caspase 4 (Figure 17).

### 2.7. DA Effects on Calcium Homeostasis

Mitochondrial viability and calpain activity assays were utilized to determine the role of calcium homeostasis in DA-induced cytotoxicity. MTT assays were performed on HK-2 cells pretreated for 45 min with EGTA (extracellular calcium chelator), BAPTA-AM (intracellular calcium chelator), or 2-APB (inositol triphosphate receptor antagonist) prior to exposure to DA. EGTA or BAPTA-AM concentrations selected for these studies were not cytotoxic based on the MTT assay results. Higher concentrations of BAPTA-AM induced some alterations in the MTT assay and were omitted from any studies with DA. Pretreatment with the intracellular calcium chelator, BAPTA-AM, completely protected against DA-induced cytotoxicity at 2, 8, and 24 h as seen by MTT assays (Figure 18). Within 2 h, neither EGTA or 2-APB showed any statistically significant protection, however, both showed partial protection within 8 h at 30 mg I/mL (*p* < 0.05) when compared to the DA group (Figure 18). EGTA and 2-APB provided partial protection from DA-induced cytotoxicity within 24 h exposure to DA (*p* < 0.05) when compared to the DA group (Figure 18). Calpain activity assays were performed to explore the role of calcium dependent proteases in DA-induced apoptosis. HK-2 cells exposed to 30 mg I/mL for 24 h experienced a two-fold increase in calpain activity (*p* < 0.05) compared to vehicle control (Figure 19). However, pretreatment with 10 µM BAPTA-AM or calpeptin, a calpain inhibitor, prior to DA exposure completely inhibited calpain activity (Figure 19). 

To investigate the role of DA-induced calcium dysregulation in mitophagy, oxidative stress, and apoptosis, protein expression of LC3BI and II, protein carbonylation, 4HNE, and caspase 12 were measured via Western blots and Oxyblot. HK-2 cells pretreated with BAPTA-AM completely abrogated the conversion of LC3BI to LC3BII (*p* < 0.05) induced by exposure to DA (Figure 20). Pretreatment with the calpain inhibitor partially inhibited the initiation of mitophagy (*p* < 0.05) indicating that calpain activation may play a role in mitochondrial turnover (Figure 20). Chelation of intracellular calcium or inhibition of calpain activation decreased protein carbonylation, however, not to a significant degree (Figure 21). However, DA-induced 4HNE adduct formation was completely abrogated (*p* < 0.05) in response to BAPTA-AM and calpeptin (Figure 21). Caspase 12 activation was also attenuated in response to BAPTA-AM and calpeptin (*p* < 0.05) (Figure 22) indicating that caspase 12 activity is not dependent on ER stress but can be activated in response to calpain activity. This conclusion was further supported by the finding of no change by DA in GRP78 and CHOP (Figure 9 and Figure 10).

## 3. Discussion

The use of RCM in radiographic imaging procedures does not show any sign of slowing down. Over 75 million contrast-requiring procedures are performed every year worldwide, and since 2006, the number of CT scans has increased by over 800% [2,19]. As patient life expectancy increases, the need for diagnostic procedures such as percutaneous coronary intervention (PCI) and cardiac catheterizations linearly increase as well [20]. Unfortunately, the combination of comorbidities that decrease renal function and exposure to RCM leads to CI-AKI resulting in increased length of hospital stay, cardiovascular events, end-stage renal disease, and all-cause mortality [21,22,23]. RCM toxicity is associated with severe renal vasoconstriction, activation of multiple inflammatory pathways, and direct tubule damage. The combination of these humoral responses can induce renal injury ranging from non-symptomatic increases in SCr to extensive damage resulting in permanent kidney failure and the need for dialysis. Although the effects of RCM exposure have been well documented, the exact mechanisms of toxicity have not been fully elucidated. A vast number of clinical reports, observational studies, and in vivo studies have explored the incidence of CI-AKI following exposure to RCM, most finding significant risk; however, little is known about the initial sources of renal injury and direct cellular toxicity pertaining to RCM exposure. Discovering and understanding the mechanisms of toxicity is imperative for the development of an appropriate preventative measure or treatment option to mitigate CI-AKI. 

To determine the source of RCM induced cytotoxicity, in vitro models must be implemented to eliminate renal hemodynamic and inflammatory responses to RCM exposure. Various cell models have been utilized throughout the years including MDCK, LLC-PK1, HEK-293, and NRK-E52, however, mechanistic studies require a model that can be translated directly to human physiology. Therefore, the human kidney cell model, HK-2, was chosen. In order to determine the first onslaught of cellular toxicity, the model in question must be exposed to concentrations of RCM that the kidney would experience in a clinical setting. On average, the adult human male, aged 19–95 years, have blood plasma volumes of 45.2 to 53 mL/kg body weight [24]. An effective dose of an RCM for CT imaging such a coronary arteriography can range from 45 to 150 mL. Therefore, for a 75 kg adult male, blood plasma levels of RCM will range from 4 to 15 mg I/mL [25]. These concentrations can effectively be doubled for larger patients and dosages can continue to increase until the effective termination limit of 300 mL of a 76% DA preparation or equivalent has been injected, approximately 30 mg I/mL [26]. It is not uncommon for mechanistic studies to expose in vitro models to concentrations of RCM over 150 mg I/mL, which is up to twenty-five-fold greater than blood plasma levels. In this current study, a wide range of DA concentrations (0, 2, 5, 10, 15, 18, 23, 28, and 30 mg I/mL) were chosen to encompass what the kidney would experience following a modern imaging procedure. Although an extensive number of peer-reviewed studies indicate the roles of various pathways and propose multiple mechanisms of toxicity, the initial source of damage has yet to be identified at clinically relevant concentrations. 

Our study has been the first to show that clinically relevant concentrations of DA induces cytotoxicity. MTT conversion was used as a measure of mitochondrial viability because the reduction of the yellow tetrazolium dye to its purple formazan counterpart is performed primarily by mitochondrial dehydrogenases [27]. Mitochondrial viability was decreased in HK-2 cells within 2 h at 15 mg I/mL and continued to decrease at 8 and 24 h at 2 mg I/mL as shown by MTT to formazan conversion. It is important to mention that a decrease in MTT conversion does not necessarily demonstrate a decrease in the total number of viable cells and vice versa [28]. Hence, trypan blue exclusion assays were performed to verify that a decrease in MTT conversion was due to a decrease in mitochondrial viability and not a decrease in total number of viable cells. Healthy cells with an intact cellular membrane will exclude the trypan blue dye, whereas dead cells will allow the dye to leak into the cytosol [29]. Unlike the results obtained from the MTT assays, a decrease in cell viability was not apparent until HK-2 cells were exposed to at least 23 mg I/mL for 24 h. This indicates that HK-2 cells experience a statistically significant decrease in mitochondrial viability at a much lower concentration and at an earlier time point when compared to the noticeable decrease in cell viability. 

The mitochondrial density of the kidney is relatively high compared to the rest of the body, second only in mitochondrial content and oxygen consumption to the heart [30]. The high mitochondrial content is due to the PT being responsible for the majority of transport within the kidney accounting for approximately 80% of all transport in the kidney [31]. Consequently, PT cells are fairly resistant to induction of pro-apoptotic mechanisms via multiple mitochondrial protection pathways. PT cells experiencing intracellular stressors such as ischemia, reaction to xenobiotics, or inflammatory responses will undergo mitochondrial swelling, fission, and mitophagy in an attempt to withstand the insult [32]. HK-2 cells have similar activity to in situ PT cells [15,17] and this could explain the discrepancy between the values obtained from the MTT and trypan blue exclusion assays.

The effects of RCM on mitochondrial function have been briefly studied in various models. Basal and uncoupled respiration measured using a Clark oxygen electrode were decreased in isolated PTs of New Zealand white rabbits after exposure to DA, ioxaglate, and iopamidol [33,34,35]. It has also been shown that exposing HK-2 or LLC-PK1 cells to ioversol, iodixanol, or iohexol induces an increase in mitochondrial ROS production, as well as, depolarization of mitochondrial membranes and stimulated the release of cytochrome c [7,36]. ATP production, and complex I and complex III activity were decreased in male Wistar albino rats in response to exposure to DA [37]. The combination of these previous studies indicates that there is an interaction between the mitochondria of PT cells and RCM, however, the source of this dysfunction still eludes us. 

This was the first study to conduct real-time, high throughput screening to determine the effects of DA on the mitochondria of live, intact PT cells. Using the Seahorse XFe96 Analyzer, the effects of clinically relevant concentrations of DA on mitochondrial respiration, glycolysis, ATP production, and mitochondrial fuel utilization were determined in HK-2 cells. Exposure to DA induced statistically significant decreases in basal and maximal respiration, spare respiratory capacity, and ATP production within 8 h and continued to decrease these parameters through 24 h as shown by the cell mito stress test results (Figure 3). Unfortunately, a global decrease in mitochondrial respiration does not necessarily predict the source of mitochondrial damage; for instance, the environmental agents 2,2′-methylenebis(4-chlorophenol) and pentachlorophenol were determined to be uncoupling agents because the compounds reversed oligomycin-induced inhibition of OCR [38]. Previous work in our lab demonstrated that the HIV medication tenofovir may be an ATP synthase inhibitor due to its reduction of OCR and no effect on spare respiratory capacity in HK-2 cells. A study performed by Namba et al. determined that renal PTs experiencing metabolic acidosis in response to ammonium chloride showed similar results to HK-2 cells exposed to DA [39]. The Namba group concluded that the global decrease in respiration was due to an increase in mitophagy. The role of mitophagy in CI-AKI will be discussed in more depth later. 

The pesticide maneb has been shown to decrease glycolysis and glycolytic reserve and increase OCR indicating that it differentially affects glycolysis and stimulates mitochondrial respiration [40]. Comparable to neuroblasts treated with maneb, HK-2 cells that are exposed to DA demonstrate decreases in both glycolysis and glycolytic reserve as shown by the cell glyco stress tests (Figure 4), however, this interpretation must be approached with caution. HK-2 cells depend on oxidative phosphorylation to a much greater extent than glycolysis as a source of energy production. This is evident by the very slight increase in ECAR following the addition of oligomycin, indicating that the spare glycolytic capacity of HK-2 is insignificant relative to oxidative phosphorylation. It is possible that the decrease in glycolysis was not due to damage to glycolytic machinery but, more likely, the disappearance of glycolytic substrates in response to the cells attempting to maintain ATP levels in the absence of functioning mitochondria. 

In order to verify that exposure to DA results in mitochondrial dysfunction and the reduction in glycolysis and glycolytic capacity was due to downstream effects, the real-time ATP rate assay measured total ATP, mitochondria-linked ATP, and glycolysis-linked ATP production. In response to exposure to DA, HK-2 cells experienced a statistically significant decrease in total ATP and mitochondrial ATP production. Conversely, a decrease in glycolytic ATP production was apparent, however, not to a significant degree. These results indicate that exposure to DA affects the mitochondria preferentially to the glycolytic pathway and the decreases that were being observed were in response to energy demand or downstream effects of mitochondrial dysfunction. 

The ability of the mitochondria to utilize available fuel sources is vital for maintaining energy homeostasis. The mito fuel flex test determines the rate of oxidation of the three major mitochondrial fuels: pyruvate, glutamine, and long-chain fatty acids (LCFA). The assay determines the reliance on a particular pathway to maintain basal respiration, labeled fuel dependency, followed by the overall fuel capacity of each fuel source. The difference of these two values is the mitochondrial fuel flexibility, or the mitochondria’s ability to compensate for an inhibited pathway by using the other two pathways. UK5099 is an inhibitor of the mitochondrial pyruvate carrier (MPC) thereby blocking the glucose oxidation pathway and determining the dependency and flexibility of the mitochondria to utilize pyruvate produced via glycolysis. BPTES is an allosteric inhibitor of glutaminase (GLS1) resulting in inhibition of the glutamine oxidation pathway. GLS1 converts glutamine to glutamate which is then converted to α-ketoglutarate by glutamate dehydrogenase to be used in the citric acid cycle. Finally, etomoxir inhibits the LCFA transporter carnitine–palmitoyl transferase (CPT1A) which is critical for translocating LCFA from the cytosol into the mitochondrial matrix to be used for β-oxidation. 

In response to exposure to clinically relevant concentrations of DA, HK-2 cells ability to oxidize glucose, glutamine, and fatty acid was not affected to a significant degree. The flexibility of glucose and glutamine oxidation slightly increased in response to low and intermediate concentrations of DA; however, the increase also did not reach levels of significance. Aerobic respiration is the primary mechanism of ATP production in PT cells, with the central source of ATP coming from β-oxidation of LCFA, such as palmitate [30]. According to the results of the mito fuel flex test, the oxygen consumption linked to each oxidative pathway within the mitochondria were approximately equivalent, indicating that HK-2 cells depend on each of the three pathways to a similar degree. It is also important to note that although we saw a decrease in every parameter of mitochondrial function as shown by the cell mito stress tests, the ability of the mitochondria to utilize the different fuel sources were not diminished. One would expect there to be a decrease in at least one oxidative pathway as a potential source of damage within the mitochondria; for example, if there was a decrease in glutamine oxidation in response to exposure to DA, it could be concluded that the diminishment of basal and maximal respiration seen in the cell mito stress tests could be attributed to dysfunction of electron transport chain machinery, glutamine conversion to α-ketoglutarate, or transport of glutamine into the mitochondria. The ability of the HK-2 cells to utilize LCFA was very slightly decreased in response to DA, and this could be explained by accumulation of LCFA as triglycerides within the cytosol. Triglycerides have been shown to accumulate within the cytosol of injured HK-2 cells, thereby decreasing the available LCFA to be utilized as a fuel source [41]. It is possible that the damage induced by DA at 8 h was not severe enough to induce a decrease in the ability of the mitochondria to utilize each of the fuel sources, but we can conclude that the observed decrease in mitochondrial function was not due to diminishment of the ability of the mitochondria to utilize the three major fuel sources.

A decrease in observable mitochondrial function in the absence of noticeable impairment of the oxidative machinery or transport of fuel may not indicate direct mitochondrial damage but could have been caused by a decrease in the overall number of active mitochondria. Mitochondrial turnover, or mitophagy, is the main source of mitochondrial degeneration and has been implicated in a number of diseases and conditions including CI-AKI [36,42,43]. The conversion of LC3BI, soluble form, to LC3BII, lipid form, is considered the major determining factor of mitophagy via the direct interaction of LC3 adapters and mitochondrial substrates [44]. HK-2 cells exposed to 200 mg I/mL of iohexol and iodixanol expressed a statistically different increase in the ratio of LC3BII/LC3BI and an increase in p62 expression demonstrating an increase in mitophagy [36]. In our study, clinically relevant concentrations of DA induced a significant increase in the ratio of LC3BII/LC3BI within 8 h of exposure indicating that the diminishment of cellular respiration may not be due to mitochondrial damage but mitophagy. Our studies also suggest that mitophagy may occur at levels that exist in the PT of a patient during development of CI-AKI. It has been shown that mitophagy may play a role in protecting PT cells from CI-AKI. Pretreatment of HK-2 cells with the inhibitor of autophagy, 3-methyladenine, resulted in more severe cellular toxicity induced by RCM [36]. The increase of mitophagy seen in this study potentially explains why conversion of MTT was diminished within 2 h of DA exposure but cell viability was not decreased until 24 h. 

Disturbances in cellular homeostasis such as alterations in available ATP, redox status, and calcium regulation will result in misfolding or unfolding of proteins within the ER. Accumulation of misfolded or unfolded proteins within the ER activates the UPR and, if the instability is prolonged, will result in ER stress and apoptosis [45]. ER stress has been shown to play a role in a large number of pathophysiological disorders including the pathogenesis of ischemia/reperfusion injury and CI-AKI [46]. The role of ER stress has been established in HK-2 and NRK-E52 cells exposed to HOCM and LOCM has previously been established. NRK-E52 cells exposed to 60 mg I/mL DA or 100 mg I/mL iopromide resulted in significant increases in GRP78, phosphorylated RNA-dependent protein kinase-like ER kinase (PERK), phosphorylated eukaryotic translation initiation factor 2α (eIF2α), and CHOP expression [47,48]. HK-2 cells treated with 40 mg I/mL of DA induced statistically significant increases in GRP78, ATF4, CHOP, and caspase 12 mRNA levels [49]. However, studies in our laboratory concluded that exposing HK-2 cells with clinically relevant concentrations of DA could not induce significant increases in GRP78 expression or the downstream pro-apoptotic protein CHOP. These findings indicate that activation of UPR or ER stress does not play a role in the decrease in cell viability in response to DA exposure.

Interestingly, we found that exposure to 28 and 30 mg I/mL induced a significant increase in caspase 12 expression indicating activation of the ER transmembrane bound protease via another mechanism. Although the mechanism of activation of caspase 12 is not entirely understood, the role of cytokines and the calcium dependent class of proteases called calpains may play a role [50]. The levels of calcium within physiological systems is tightly regulated. This is due to the fact that calcium acts as an essential intracellular and extracellular messenger in numerous cellular events such as hormone secretion, muscle contraction, immune responses, activation of neuronal networks, cell survival, and cell death. Maintenance of extracellular calcium is held in a narrow range of 8.5–10.5 mg/dL and maintaining this range is important for intracellular calcium homeostasis. 

Calcium overload within the mitochondria is considered to be one of the main driving factors of cell death. Mitochondrial membrane permeability transition (mPT) is considered the initiator of the intrinsic apoptotic pathway and induction involves numerous factors. The only dogma pertaining to mPT is the accumulation of calcium within the mitochondrial matrix. Inhibiting calcium influx into the matrix with ruthenium red completely abrogates mPT verifying the role of calcium in mPT [51]. Accumulation of large amounts of calcium within the matrix leads to the opening of a large channel within the mitochondrial inner membrane referred to as the mitochondrial membrane permeability transition pore (mPTP). As the mPTP opens, proteins (<1.5 kD) and solutes flood into the matrix resulting in uncoupling of oxidative phosphorylation, depletion of ATP, and permeabilization of the outer mitochondrial membrane [52,53]. Mitochondrial outer membrane permeability results in leakage of cytochrome c which binds to apoptotic protease activating factor (APAF1) to form the apoptosome. The apoptosome continues to activate caspase 9 and induce apoptosis [54].

The role of calcium homeostasis in the pathogenesis of CI-AKI has been established previously by various labs. Intracellular calcium overload may play a role in ROS overproduction, p38 MAPK activation, and apoptosis in CI-AKI [55,56]. Another study determined that exposure to 150 mg I/mL ioversol for 24 h induced cytochrome c release and activation of caspase 3 in glomerular endothelial cells isolated from adult male Sprague–Dawley rats, however, pretreatment with the intracellular calcium chelator BAPTA-AM (5 µmol/L) completely attenuated the cytotoxicity [57]. Our studies confirm the Zhao group findings as pretreatment with BAPTA-AM completely alleviated DA-induced cytotoxicity in HK-2 cells as seen by MTT conversion. Whereas the Zhao study found that the extracellular calcium chelator EGTA had no effect on protecting glomerular endothelial cells from RCM-induced cytotoxicity, pretreating HK-2 cells with EGTA did afford partial protection from DA-induced cytotoxicity. Additionally, pretreatment of HK-2 cells with the inositol triphosphate receptor (IP_3_R) antagonist 2-APB also provided partial protection from diminished mitochondrial viability in response to DA exposure. Pretreatment with BAPTA-AM also completely abrogated the conversion of LC3BI to LC3BII in response to DA further proving that DA does cause intracellular calcium overload and mitophagy. It is clear from these results that calcium plays a major role in mitochondrial viability, however, the source of this calcium is still in question. It is likely that the DA-induced increase in intracellular calcium was caused by a combination of the release of calcium from the ER and calcium influx through store-operated calcium channels (SOCs) due to the fact that both EGTA and 2-APB provided partial protection.

Aside from inducing mPT, intracellular calcium overload can also induce cellular damage by other mechanisms. Prolonged surges in intracellular calcium has long been thought to activate a class of calcium-activated non-lysosomal cysteine proteases called calpains [58]. It is hypothesized that during events of high intracellular calcium, inactive calpain translocates from the cytosol to the ER membrane and autocatalyzes resulting in the dissociation of the 30 kD active subunit [59]. Calpain substrates do not have a specific recognizable amino acid sequence, therefore, a large variety of proteins are targets [60] including other pro-apoptotic enzymes like caspase 12. In fact, it has been previously shown that activation of caspase 12 requires calpain activation in vivo [61]. It has been shown, that pretreatment with calpain or calpain inhibitor-1 prior to exposure to RCM results in decreased antioxidant activity or reduced renal impairment, respectively, in rats [62,63,64]. Our studies have shown that HK-2 cells exposed DA for 24 h induced a two-fold increase in calpain activity and inhibiting calpain activity with calpeptin or chelating intracellular calcium with BAPTA-AM completely abrogated DA-induced calpain activation. To verify the role of caspase 12 activation by calpain activity, HK-2 cells were pretreated with BAPTA-AM or the calpain inhibitor calpeptin prior to DA exposure. As a result, both BAPTA-AM and calpeptin inhibited the DA-induced increase in caspase 12 expression indicating that caspase 12 activation takes place in response to intracellular calcium overload and not to ER stress. 

Perhaps the most studied topic pertaining to the cytotoxicity of RCM is oxidative stress. As the production of reactive oxygen species (ROS) overwhelms the ability of a cell’s innate antioxidant systems to repress ROS reactivity, the ROS continuously and cumulatively damage the cell via interactions with cellular proteins, DNA, and other cellular structures [65]. Sources of ROS production vary but the most common endogenous sources are as byproducts of cellular metabolism through the electron transport chain (ETC), specifically complexes I and III, one-electron reduction of O_2_ through enzymatic catalysis by NADPH oxidase (NOX) or xanthine oxidase (XO), or uncoupling of endothelial nitric oxide synthase (eNOS) [66]. The mitochondria ETC and NOX are by and large the major sources of ROS generation [67,68]. In response to ROS production, cells have innate antioxidant systems to reduce oxidants to water and O_2_. The most common players in the endogenous antioxidant systems include: the enzymatic ROS scavengers superoxide dismutase (SOD), catalase (CAT), glutathione peroxidase (GPx), and glutathione transferase, and the non-enzymatic scavengers glutathione (GSH) [69]. In the context of CI-AKI, a plethora of studies have been performed to confirm the formation of free radicals and oxidative stress as a result of exposure to RCM [5,70,71,72]; however, very few studies have addressed the source of ROS production. The purpose of this study was to verify that clinically relevant concentrations do induce an increase in ROS and oxidative stress and identify the source of the excessive generation of ROS. 

Protein carbonylation is a generic term used to describe the irreversible interaction between reactive ketones and aldehydes formed in the presence of ROS and the side chains of proteins, specifically lysine, arginine, proline, and threonine [73]. This reactive carbonyl moiety will further impede biomolecule function, cause inflammation, cell toxicity, and induce apoptosis. Accumulation of proteins with carbonylated side chains indicates an increase in oxidative stress. Protein carbonyls can be recognized via derivatization with 2,4-dinitrophenylhydrazine (DNPH) to form hydrazones that can detected using antibodies against DNPH-derivatized proteins [74]. A more specific biomarker for oxidative stress is the formation of protein-bound carbonyls induced by excessive lipid peroxidation. Most lipid peroxidation products are strong electrophiles and readily form Michael adducts with lysine, cysteine, and histidine side chains [75]. The primary α,β-unsaturated hydroxalkenal formed during lipid peroxidation, 4HNE, is one of the most commonly used biomarkers for oxidative stress and can easily be detected via immunoblotting. 

Our studies indicated that exposure to clinically relevant concentrations of DA induced oxidative stress in HK-2 cells as shown by an increase in protein carbonylation and 4-hydroxyl-nonenal (4HNE) protein adduct formation within 24 h, however, no increase in oxidative stress was apparent within 8 h. It is hypothesized that a major portion of direct cytotoxicity induced by RCM is caused by oxidative stress, however, our results indicate mitochondrial turnover at an earlier time point. Therefore, at clinically relevant concentrations, the increase in oxidative stress is not the initial or sole cause of HK-2 cytotoxicity but is due to damage that is already taking place within the cell. 

The major source of endogenous ROS production is complex I and complex III of the electron transport chain within the mitochondria. Under normal physiological conditions, ROS generated in the mitochondria are involved in cellular crosstalk, signal integration of cell proliferation, differentiation, inflammation, repair, and apoptotic pathways [76]. However, in circumstances of perturbances in cellular homeostasis, such as alterations in NADH/NAD+ ratio, metabolic disturbances, or mitochondrial membrane damage, mitochondrial ROS generation surges [77]. Within the mitochondria of an affected cell, electrons leak from the intermembrane space at complex I and complex III where it interacts with O_2_ to yield superoxide (O_2_^·−^) radical. The mitochondria have an internal system to dispose of excess ROS that includes manganese superoxide dismutase (MnSOD) which dismutates O_2_^·−^ into H_2_O_2_ and glutathione peroxidase (GPx) which fully reduces H_2_O_2_ into water [77]. The mitochondria are at a disadvantage, however, due to the fact that it lacks catalase, a H_2_O_2_ reducing enzyme, and depends on the reduction of glutathione disulfide to glutathione via glutathione reductase (GR) to reduce H_2_O_2_. As concentrations of the reducing equivalent NADPH decrease and glutathione disulfide increase, the antioxidant system within the mitochondria become overwhelmed [78]. Once this takes place, ROS interactions disrupt the mitochondrial membrane potential and permeability transition is eminent. As previously mentioned, mitochondrial outer membrane permeability results in leakage of cytochrome c which binds to apoptotic protease activating factor (APAF1) to form the apoptosome. A study performed by Lei et al. indicated that exposure of HK-2 cells to 200 mg I/mL iodixanol or iohexol induced statistically significant increases in mitochondrial ROS production as shown by mitoSOX and TMRE staining [36]. Our studies indicate that exposure of HK-2 cells to clinically relevant concentrations of DA induced the opposite effect. In fact, we noticed no change in 4HNE protein adduct formation or protein carbonylation within the mitochondria and the statistically significant increases in 4HNE adduction and protein carbonylation took place solely within the cytosol. To verify these findings, SOD activity assays were performed. We discovered that there was no change in MnSOD expression or activity, and the overall decrease in SOD activity was due, entirely, to a decrease in activity of the cytosolic SOD, Cu/ZnSOD. 

The role of the cytokine tumor necrosis factor alpha (TNFα) was measured in response to DA exposure as a possible source of ROS generation. TNFα is an important cytokine in inflammation and is involved in many cellular responses including pro-survival and pro-apoptotic pathways that has been shown to be affected in response to RCM exposure [79]. TNFα has been shown to induce ROS production in a number of different pathways but most noticeably is through activation of NOX, specifically NOX4 in renal parenchymal cells [80]. TNFα induces an increase in various NADPH oxidase components such as the NOX regulatory proteins p22phox and p47phox, NADPH oxidase organizer 1 (NOXO1), as well as, NOX4 itself [81,82,83]. A study performed by Jeong et al. demonstrated that exposing HK-2 cells to 150 mg I/mL iohexol induced statistically significant increases in NOX4 within 5 min; however, the group did not verify if the increase in NOX4 induced an increase in ROS at this time point in HK-2 cells [6]. In response to clinically relevant concentrations of DA, a statistically significant increase in TNFα in cell media and decrease in TNFα in cell lysate of HK-2 cells was apparent, indicating that DA does induce a mild inflammatory response in vitro. However, NOX4 expression was unaffected in response to the increase in active TNFα. From these results, we can conclude that although TNFα was activated in response to DA exposure, the intracellular signaling was not strong enough to induce NOX4 transcription or activation. We can also conclude that TNFα and NOX4 were not playing a role in ROS generation or apoptosis in HK-2 cells. The role of calcium dysregulation and ROS production will be discussed later. 

Excessive ROS generation induced by intracellular calcium dysregulation may play a role in CI-AKI, as well. Intracellular calcium levels have been linked to ROS production in response to RCM [55,56]. Calcium induced ROS production is linked to the interactions between mitochondrial calcium and increased ATP production and ROS generation, and calcium induced activation of NOX [84]. However, our studies indicated that there was no change in NOX4 activation and mitochondrial oxidative stress in response to DA exposure. Calcium may be inducing ROS production via calpain activation. In diabetic mice and human umbilical vein endothelial cells, activation of calpain correlated with an increase in ROS production, whereas, transgenic mice over-expressing the endogenous calpain inhibitor calpastatin experienced significant reduction in ROS production [85]. Our study indicated that pretreatment with BAPTA-AM or calpeptin completely abrogated 4HNE adduct formation, however, protein carbonylation was not decreased to a significant degree. From these results, it can be concluded that the increase in lipid peroxidation that took place in response to DA exposure was linked to calcium dysregulation and calpain activity. 

Apoptosis can be induced via two main pathways: the mitochondrial, or intrinsic, pathway; the ligand-mediated, or extrinsic, pathway. Both pathways involve two types of caspases: the initiator caspases (caspase 2, 4, 8, 9, 10, 11, and 12) and the effector caspases (caspase 3, 6, 7). Activation of specific apoptotic pathways can be determined by measuring the appropriate caspase. Cleavage of caspase 9, for example, occurs in response to activation of the apoptosome and can be used as a determinant of initiation of the intrinsic apoptotic pathway. The apoptosome will then interact with and cleave executioner caspases such as caspase 3 resulting in programmed cell death. As previously mentioned, exposure to DA induces an increase in expression of the initiator caspase, caspase 12; however, caspase 12 activation occurs in the absence of initiation of the UPR or ER stress. Caspase 4 was also measured as an indicator of the extrinsic apoptotic pathway. Caspase 4 is in response to binding of TNFα to the TNFα receptor 1 (TNFR1), activation of the TNFα receptor-associated death domain (TRADD), and induction of extrinsic apoptosis. Caspase 4 has also been linked to ER stress linked apoptosis. HK-2 cells exposed to clinically relevant concentration of DA for 24 h did notice a statistical increase in TNFα activation but there was no noticeable change in caspase 4, indicating that ER stress and the extrinsic apoptotic pathway did not play roles in DA-induced apoptosis. Cleavage of caspase 3 was statistically increased in response to clinically relevant concentrations of DA, indicating that apoptosis was occurring within 24 h and mitochondrial dysfunction may have been the major contributing factor to the decrease in cell viability as seen by leakage of cytochrome c and the increase in mitophagy. 

## 4. Methods

### 4.1. Chemicals and Reagents

DA (S4506-50G), MTT (M2128-10G), and SOD activity kit (19160) were purchased from Sigma-Aldrich (St. Louis, MO) and was used for all studies. The vehicle used for cell treatments was phosphate buffered saline (PBS) purchased from Fisher Scientific (Gibco, Pittsburg, PA, USA, Item No. 14175-095). All other chemicals were of the highest quality and procured from Sigma-Aldrich or Fisher Scientific Inc. Antibodies were purchased as indicated in the sections below. The OxyBlot™ Protein Oxidation Detection Kit was purchased from EMD Millipore (Burlington, MA, USA, Item No. S7150).

### 4.2. Cell Line and Diatrizoic Acid (DA) Treatment

Non-cancerous human immortalized kidney epithelial cells (HK-2) were purchased from the American Type Culture Collection (ATCC, Manassas, VA, USA, Item No. CRL-2190) and were cultured according to ATCC guidelines. Cells were grown in keratinocyte-free media with added bovine pituitary extract (50 μg/mL) and recombinant epithelial growth factor (5 ng/mL) purchased from Fisher Scientific (Gibco, Carlsbad, CA, USA, Item No. 17005-042). Cells were grown in a warm, humidified incubator set to 37 °C with 5% CO_2_. HK-2 cells were plated into six-well cell culture plates (Corning, Sigma Aldrich Item No. CLS3516) at a cellular density of 750,000 cells/mL and allowed to grow for 48 h prior. Media was subsequently replaced and cells were treated with a final concentration of 0, 2, 5, 10, 15, 18, 23, 28, or 30 mg I/mL DA for 2, 8, or 24 h prior to cell lysate collection. Vehicle control was an equal volume of PBS. Following the treatment period, media was removed via aspiration and cells were collected using trypsin–EDTA (0.25%) (Gibco, Carlsbad, CA, USA, Item No. 25200-072) for sample analysis. 

### 4.3. Mitochondrial and Cell Viability

HK-2 cells were plated into 96-well cell culture plates (Cyto One Scientific, Ocala, FL, USA, Item No. CC7682-7596) at a cellular density of 37,500 cells/mL and allowed to grow for 48 h. Following the equilibration period, media was replaced, and cells were treated with a final concentration of 0, 2, 5, 10, 15, 18, 23, 28, or 30 mg I/mL DA for 2, 8, or 24 h. Vehicle control was an equal volume of PBS. Mitochondrial viability was assessed using the MTT assay [86]. The MTT assay relies on the conversion of tetrazolium dye 3-(4,5-dimethylthiazol-2-yl)-2,5-diphenyltetrazolium bromide (MTT) (Sigma Aldrich, St. Louis, Mo, USA, Item No. M5655-5X1G) to formazan by mitochondrial oxidoreductases. 

Trypan blue exclusion was used as a confirmation that the DA mediated decline in MTT reduction was due to a decrease in mitochondrial viability and not an interaction between DA and mitochondrial reductase enzymes. The trypan blue exclusion assay was run using the Countess II FL cell counter (Thermo Fisher Scientific Inc., Pittsburgh, PA, USA) An aliquot of collected cells were diluted 1:1 with 40% *w/v* trypan blue solution (Sigma-Aldrich, Item No. T6146). The solution was lightly mixed via pipetting and a 10 μL aliquot was transferred to the Cell Countess II FL reusable glass slide prior to inserting the slide into the instruments sample port. The Cell Countess II FL measures total cells, living cells, and dead cells as a measure of overall cell viability. 

### 4.4. Mitochondrial Isolation

Mitochondria were isolated from HK-2 cells collected with trypsin–EDTA using a Mitochondrial Isolation Kit for Cultured Cells (Thermo Scientific, Pittsburgh, PA, USA Item No. 89874) and differential centrifugation. Mitochondria were isolated according to the manufacturer’s directions, with the exception of added protease inhibitor (Thermo Scientific, Item No. 78430). Briefly, cells were centrifuged at 2000× *g* for 10 min at 4 °C and resuspended in 800 μL of Reagent A with added protease inhibitor (10 μL/mL). Samples were vortexed for five sat maximum speed and then incubated on ice for 5 min. Following addition of 10 μL of Reagent B, samples were incubated on ice and vortexed for 5 s at maximum speed every min for 10 min. Next, 800 μL of Reagent C with added protease inhibitor (10 μL/mL) was added, inverted multiple times to mix, and the samples were centrifuged at 700× *g* for twenty min at 4 °C. The supernatant was transferred to new microcentrifuge tubes and centrifuged at 12,000× *g* for 15 min at 4 °C. The supernatant (cytosolic fraction) was transferred to a new microcentrifuge tube and the pellet (mitochondrial fraction) was resuspended in 100 μL cell lysis buffer. The cellular and mitochondrial fractions were then placed in a –80 °C freezer prior to protein quantification via Bradford assay. 

### 4.5. Oxyblot and Western Blot

Western blot analysis was conducted to assess the expression of glucose-related protein (GRP78), C/EPB homologous protein (CHOP), 4-hydroxynonenal (4HNE), manganese superoxide dismutase (MnSOD), NADPH oxidase 4 (NOX4), tumor necrosis factor-α (TNF-α), cytochrome c, caspase 3, caspase 4, and caspase 12. Protein concentration was determined using the Bradford assay [87]. An aliquot of sample containing 40 μg of protein were placed in microcentrifuge tubes and denatured by placing in boiling water for 5 min followed by separation on a 12.5% polyacrylamide gel via electrophoresis and transferred onto a 0.45 µM nitrocellulose membrane (Bio-Rad, Hercules, CA, USA, Item No. 1620115). Successful transfer and protein loading were verified using Memcode Reversible Protein Stain Kit (Pierce Biotechnology, Rockford, IL, USA, Item No. PI-24580). Membranes were blocked at room temperature using 5% *w/v* milk/TBST solution (10 mM Tris-HCl, 150 mM NaCl, 0.1% Tween-20; pH 8.0), 1% Bovine Serum Albumin (BSA)/TBST, or 5% BSA/TBST solution for 1 h. Membranes were next incubated with continual shaking overnight at 4 °C with primary antibodies: GRP78 (1:1000 dilution, Abcam Inc.; Cambridge, MA, USA, Item No. ab21685), CHOP (1:1000, Cell Signaling Technology; Danvers, MA, USA, Item No 5554), MnSOD (1:5000, Abcam Inc; Cambridge, MA, USA, Item No. ab13533), TNFα (1:1000, Abcam Inc; Cambridge, MA, USA, Item No. ab66579), NOX4 (1:1000 Abcam Inc.; Cambridge, MA, USA, Item No. ab60940), LC3B (1:1000, Abcam Inc; Cambridge, MA, USA, Item No. ab48394), 4HNE (1:1000 dilution, Calbiochem; Merck, Darmstadt, Germany, Item No. 393207), cytochrome c (1:1000, Santa Cruz Biotechnology, Santa Cruz, CA, USA, Item No. sc-7159), caspase 3 (1:1000, Cell Signaling Technology; Danvers, MA, USA, Item No. 9662), caspase 4 (1:1000, Cell Signaling Technology; Danvers, MA, USA, Item No. 4450), and caspase 12 (1:1000, Cell Signaling Technology; Danvers, MA, USA, Item No. 2202) in blocking solution. The membranes were washed three times with TBST or PBST for 10 min each followed by incubation with goat anti-rabbit HRP-linked secondary antibodies diluted to 1:2000 in blocking solution for 1 to 1.5 h. Membranes were washed again with TBST or PBST and then developed using Amersham ECL Western Blotting Detection Agent (GE Healthcare Life Sciences, Marlborough, MA, USA, Item No. RPN2232). A BioRad chemic-doc system was used to capture the gel image and used for densitometry analysis (version 4.0.1, Catalog No. 170-9690, BioRad, Hercules, CA, USA).

Aside from 4HNE protein adduct formation, protein carbonylation is also a marker of oxidative stress. The accumulation of carbonyl groups introduced into cellular proteins was measured using the Oxyblot Protein Oxidation Detection Kit (EMD Millipore, Burlington, MA, USA, Item No. S7150). A 15 µg aliquot of sample was derivatized as previously described [88]. Protein carbonyl moieties on cellular proteins generated by oxidative stress were derivatized in the presence of 2,4-dinitrophenylhydrazine (DNPH) to sTable 2,4-dinitrophenylhydrazone groups. The 2,4-dinitrophenylhydrazone groups were recognized by the primary antibody (1:150 dilution in 1% BSA/PBST). A Bio-Rad chemic-doc system was used to capture the gel image and used for densitometry analysis (version 4.0.1, Bio-Rad, Hercules, CA, USA, Catalog No. 170-9690).

### 4.6. Seahorse XF Assays

The Agilent Seahorse XFe analyzer allows for real-time measurements of cellular metabolic function in cultured cells. Oxygen consumption rate (OCR) and extracellular acidification rate (ECAR) were measured to interrogate key cellular functions such as mitochondrial respiration and glycolysis. Mitochondrial function and glycolysis were measured using Agilent cell mito stress tests, cell glycolysis stress tests, mito fuel flex tests, and real-time ATP rate assays following optimization of cell number per well.

HK-2 cells were cultured in XFe Culture Miniplates (175,000 cells/mL) (Agilent Technologies, Item No. 101085-004) and allowed to grow for 48 h followed by treatment with vehicle or 5, 15, 18, 23, 28, or 30 mg I/mL DA. Prior to the assay, cells were washed with assay media (Agilent Technologies, Item No. 103575-100) supplemented with 1 mM glucose (Agilent Technologies, Item No. 103577-100), 1 mM pyruvate (Agilent Technologies, Item No. 103578-100), and 2 mM glutamine (Agilent Technologies, Santa Clara, CA, Item No. 103579-100) and equilibrated in 175 or 180 μL pre-warmed assay media at 37 °C with no CO_2_ for 45 min.

### 4.7. TNFα in Cell Media and Cell Lysate

TNF-alpha is expressed on the surface of renal proximal tubular cells and is activated as part of the inflammatory response within the kidney. TNFα concentrations were measured in cell culture media using an ELISA assay kit (Abcam, Cambridge, MA, USA, Item No. ab181421) per the manufacturer’s instructions. Briefly, 50 μL of collected media and a capture/detector antibody cocktail were added to precoated wells and incubated for 1 h, shaking at 400 rpm. Following the immunocapture incubation period, the wells were washed and 3,3*t*,5,5*t*-Tetramethylbenzidine (TMB) substrate was added, producing a color change based on the amount of bound TNFα, which was then read at 450 nm. The TNFα concentration was determined using a standard curve. TNFα expression in DA treated cell lysate was determined using Western blot as described above. Each lane was loaded with 40 μg of protein; membranes were probed using a rabbit-polyclonal HRP-linked antibody for TNFα diluted to 1:1000 in 5% BSA/TBST (Abcam, Item No. ab66579). TNFα was normalized to protein and compared relative to control.

### 4.8. SOD Activity Assay

SOD activity was determined using a Fluka designed spectrophotometric kit purchased from Sigma (St. Louis, MO, USA, Item No. 19160). Cu/Zn-SOD was inhibited by incubating the sample at room temperature with sodium diethyldithiocarbamate at a final concentration of 25 mM. The difference between total SOD and MnSOD activity was calculated as a measure of Cu/ZnSOD activity. The assay was completed according to manufacturer’s recommendations.

### 4.9. Calcium Assays

To analyze the influence of intracellular calcium concentration on mitophagy, mitochondrial viability, and apoptosis, HK-2 cells were pretreated with 10 µM of the intracellular calcium chelator, 1, 2-bis (*o*-aminophenoxy) ethane-*N*,*N*,*N’*,*N’-*tetra-acetic acid (BAPTA-AM) (Sigma-Aldrich, Item No. A1076-25MG), 1 mM of extracellular calcium chelator, ethyleneglycol-bis(β-aminoethyl)-*N*,*N*,*N′*,*N′*-tetra-acetic acid (EGTA) (Sigma-Aldrich, Item No. E3889-25G), 10 µM of the inositol trisphosphate receptor antagonist, 2-aminophenyl borinate (2-APB) (Sigma-Aldrich, Item No. D9754), or 10 µM of the calpain inhibitor, calpeptin (Sigma-Aldrich, Item No. C8999), for 45 min prior to the addition of DA. HK-2 cells were then incubated for 24 h with varying concentration of DA as described above. The calpain activity assay purchased from AnaSpec Inc. (Fremont, CA, USA, Item No. AS-72149) were used to determine the role of calcium in mitophagy and apoptosis. Calpain activity assay was performed in a 96-well plate according to manufacturer’s instructions.

### 4.10. Statistical Analysis

Values represent mean ± SEM with at least three independent experiments conducted with at least two biological replicates. Differences between groups were determined with a one-way ANOVA followed by a Holm–Sidak post-hoc test with *α* < 0.05 (GraphPad Prism, GraphPad Software Inc., San Diego, CA, USA).

## 5. Conclusions

This study provided additional insight into the mechanisms of DA-induced renal epithelial cytotoxicity. DA induced a decrease in mitochondrial (MTT) and cell viability (trypan blue exclusion) within 2 h and 24 h, respectively. An 8-h DA exposure decreased basal and maximal respiration, spare respiratory capacity, and ATP production as seen by changes in OCR and ECAR in Seahorse XFe assays. The ability of the mitochondria to utilize the three major fuel sources (glucose, glutamine, and fatty acid oxidation) was unchanged at clinically relevant DA concentrations. DA increased the LC3BII/I ratio at 8 h suggesting that DA increased mitophagy. Oxidative stress was increased beginning at 24 h indicating that oxidative stress is the result of previous cellular damage and not an initial cause. Exposure to DA induced no change in MnSOD expression or activity while Cu/ZnSOD activity was decreased in response to DA exposure. The role of TNFα in DA-induced cytotoxicity was evaluated by measuring NOX4 and caspase 4 expression but no change was detected, therefore, the source of ROS overproduction remains elusive at this time. Caspase 3 and 12 activation was evident following a 24 h exposure of HK-2 cells to clinically relevant concentrations of DA indicating that DA was inducing apoptosis. Caspase 12 expression was increased despite the fact that these concentrations of DA did not induce the UPR or ER stress as shown by GRP78 and CHOP expression. Pretreatment of HK-2 cells with calcium concentration modulators, BAPTA-AM, EGTA, and 2-APB, resulted in partial or complete protection from DA-induced mitochondrial toxicity demonstrating that calcium dysregulation plays an important role in CI-AKI. To determine if calpains play a role in caspase 12 activation, calpain activity assays were performed and determined that HK-2 cells exposed to 30 mg I/mL induced an increase in calpain activity. Pretreatment with BAPTA-AM completely abrogated the activation of calpain, caspase 12 activation, and the increase in 4HNE adduct formation in response to DA exposure. Additional studies need to be conducted to determine the sources of ROS overproduction and calcium dysregulation, as well as, the role of calcium in cellular homeostasis in response to RCM.

## Figures and Tables

**Figure 1 ijms-20-04074-f001:**
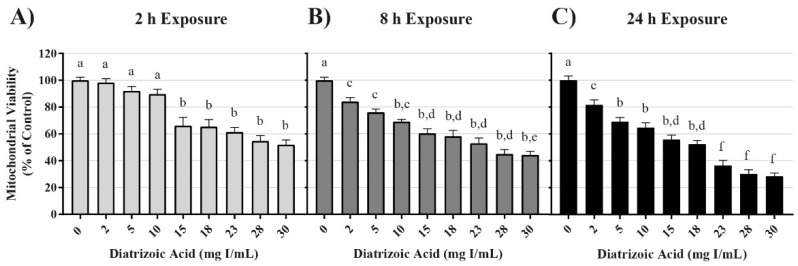
Diatrizoic acid cytotoxic effects on mitochondrial viability in HK-2 cells using 3-(4,5-dimethylthiazol-2-yl)-2,5-diphenyltetrazolium bromide (MTT). Diatrizoic acid (DA) diminished mitochondrial viability at 2 h (**A**), 8 h (**B**), and 24 h (**C**). Different letters (a–f) above each bar indicate statistical difference (*p* < 0.05) between all treatments compared across all time points (2, 8, and 24 h). Values represent mean ± SEM for three independent experiments.

**Figure 2 ijms-20-04074-f002:**
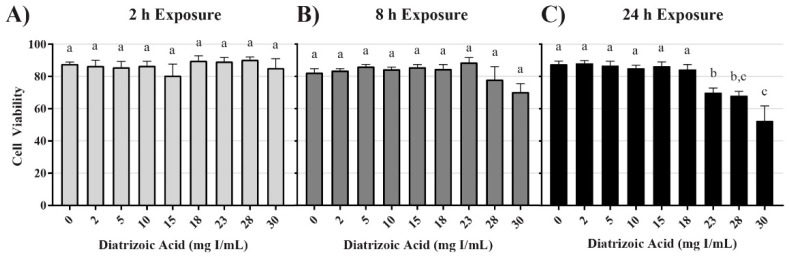
Diatrizoic acid cytotoxic effects on cell viability in HK-2 cells using trypan blue exclusion. DA diminished cell viability at 24 h (**C**) but not at 2 h (**A**) or 8 h (**B**). Different letters (a–c) above each bar indicate statistical difference (*p* < 0.05) when comparing all DA concentrations across all time points. Values represent mean ± SEM for three independent experiments.

**Figure 3 ijms-20-04074-f003:**
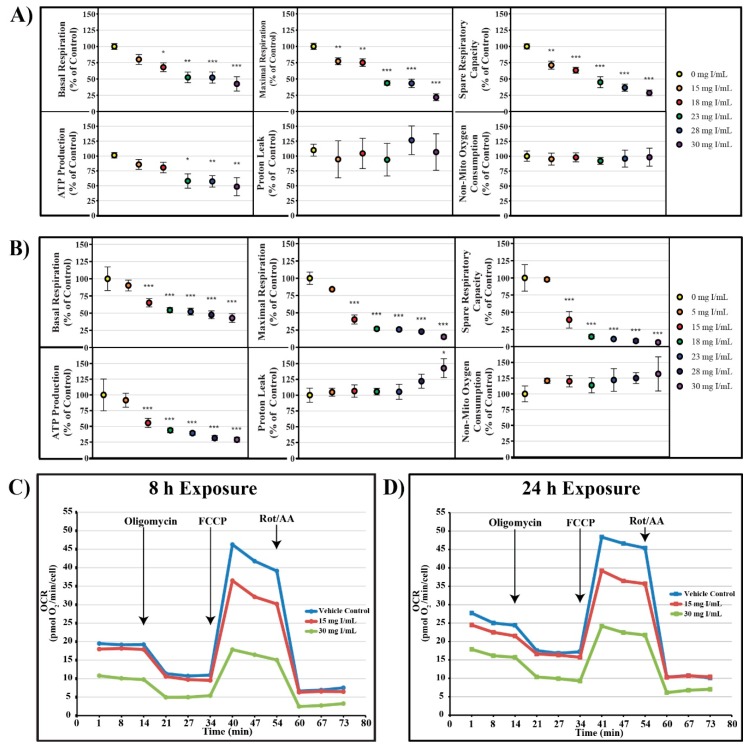
Diatrizoic acid effects on various parameters of mitochondrial respiration in HK-2 cells. DA diminished key parameters of mitochondrial respiration following 8 h (**A**) and 24 h (**B**) exposure. Representative time course profile of oxygen consumption rate (OCR) of a Seahorse cell mito stress test following exposure to DA for 8 h (**C**) and 24 h (**D**). Statistical difference from 0 mg I/mL DA depicted by an asterisk (* *p* < 0.05, ** *p* < 0.01, *** *p* < 0.001). Values represent mean ± SEM for three independent experiments.

**Figure 4 ijms-20-04074-f004:**
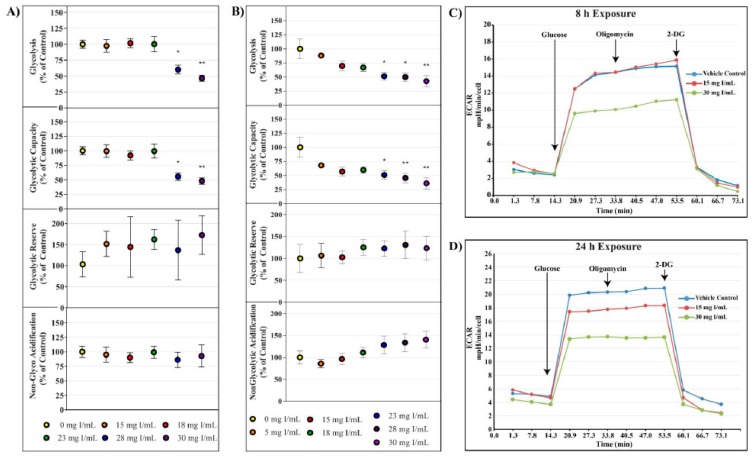
Diatrizoic acid effects on various parameters of glycolysis in HK-2 cells. DA diminished key parameters of glycolysis following 8 h (**A**) and 24 h (**B**) exposure. Representative time course profile of extracellular acidification rate (ECAR) of a Seahorse cell glycolytic stress test following exposure to DA for 8 h (**C**) and 24 h (**D**). Statistical difference from 0 mg I/mL DA depicted by an asterisk (* *p* < 0.05, ** *p* < 0.01). Values represent mean ± SEM for three independent experiments.

**Figure 5 ijms-20-04074-f005:**
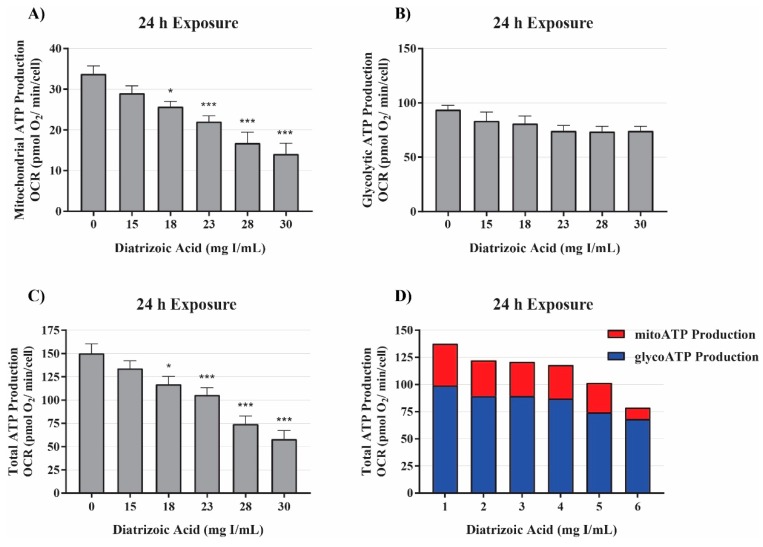
Diatrizoic acid effects on mitochondrial, glycolytic, and total ATP production. DA diminished ATP production linked to mitochondrial respiration (**A**) and total ATP production (**C**) but not glycolytic ATP production (**B**) following 24 h exposure. Representative graph of ATP production of the real-time ATP rate assay following exposure to DA for 24 h (**D**). Statistical difference from 0 mg I/mL DA depicted by an asterisk (* *p* < 0.05, *** *p* < 0.001). Values represent mean ± SEM for three independent experiments.

**Figure 6 ijms-20-04074-f006:**
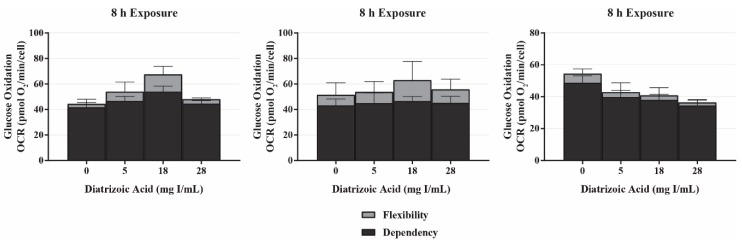
Diatrizoic acid effects on mitochondrial fuel oxidation in HK-2 cells. DA did not affect mitochondrially-linked oxidation of glucose, glutamine, and fatty acids in response to 8 h exposure. Values represent mean ± SEM for three independent experiments.

**Figure 7 ijms-20-04074-f007:**
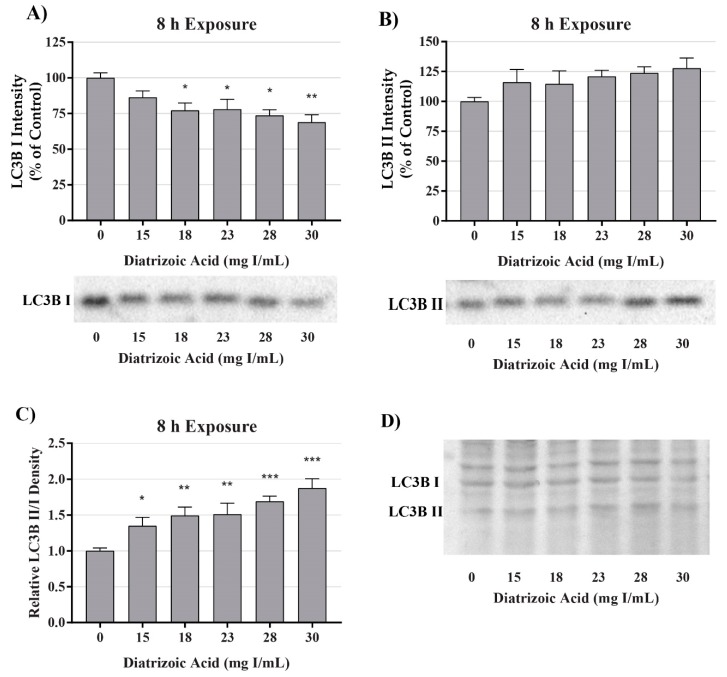
Diatrizoic acid effects on LC3B expression in HK-2 cells following 8 h exposure. DA induced mitophagy following 8 h exposure. Representative blots and cumulative densitometry included for LC3BI (**A**), LC3BII (**B**) exposure, and LC3BII/LC3B I ratio (**C**) following 8 h exposure to DA. Representative blot showing equivalent Memcode reversible stain for 40 µg loaded protein depicted for 8 h (**D**) exposure. Statistical difference from 0 mg I/mL diatrizoic acid depicted by an asterisk (* *p* < 0.05, ** *p* < 0.01, *** *p* < 0.001). Values represent mean ± SEM for three independent experiments.

**Figure 8 ijms-20-04074-f008:**
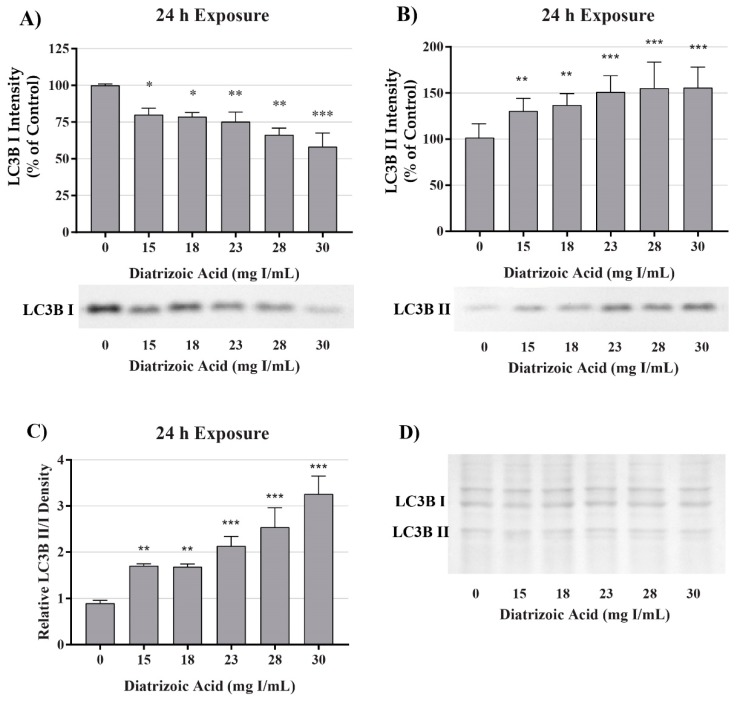
Diatrizoic acid effects on LC3B expression in HK-2 cells following 24 h exposure. DA induced mitophagy following 24 h exposure. Representative blots and cumulative densitometry included for LC3BI (**A**), LC3BII (**B**) exposure, and LC3BII/LC3BI ratio (**C**) following 24 h exposure to DA. Memcode protein staining of LC3BI and II blot loaded with 40 µg protein (**D**). Statistical difference from 0 mg I/mL diatrizoic acid depicted by an asterisk (* *p* < 0.05, ** *p* < 0.01, *** *p* < 0.001). Values represent mean ± SEM for three independent experiments.

**Figure 9 ijms-20-04074-f009:**
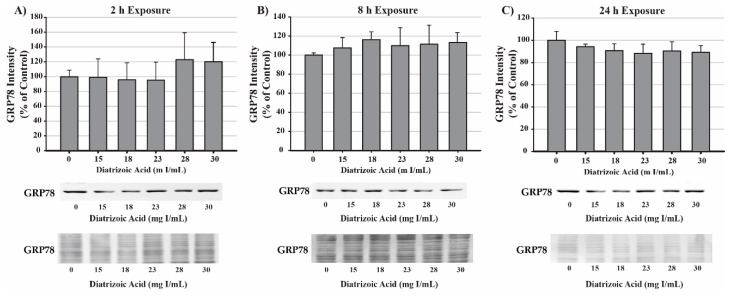
Diatrizoic acid effects on GRP78 expression in HK-2 cells. DA did not activate the unfolded protein response (UPR). Representative blots and cumulative densitometry included for glucose-regulated protein 78 (GRP78) expression following 2 h (**A**), 8 h (**B**), and 24 h (**C**) exposure to DA. Protein loading of 40 µg in each lane was visualized with Memcode reversible stain and depicted below each GRP78 blot. Values represent mean ± SEM for three independent experiments.

**Figure 10 ijms-20-04074-f010:**
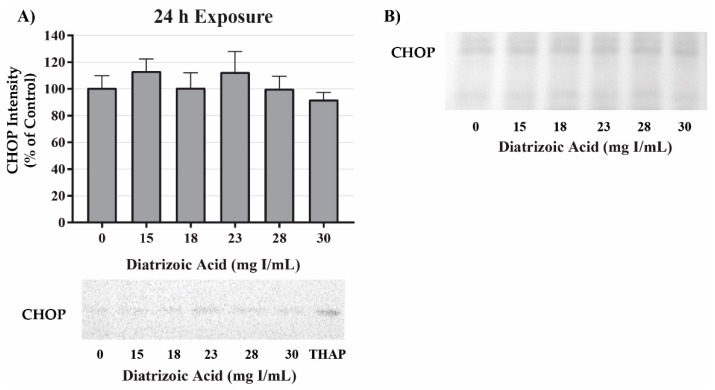
Diatrizoic acid effects on C/EBP homologous protein (CHOP) expression in HK-2 cells. DA did not induce ER stress. Representative blots and cumulative densitometry included for CHOP expression at 24 h (**A**). Panel (**B**) depicts Memcode reversible stain for 40 µg loaded protein. Positive control for CHOP expression was thapsigargin (THAP). Values represent mean ± SEM for three independent experiments.

**Figure 11 ijms-20-04074-f011:**
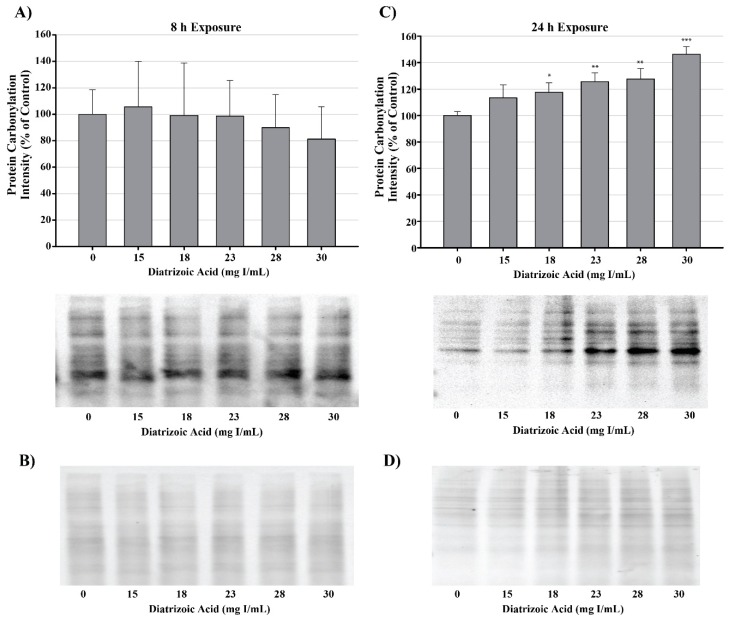
Diatrizoic acid effects on protein carbonylation in HK-2 cells. Carbonylated proteins were similar between control and DA treated cells at 8 h (**A**). DA at 18–30 mg I/mL increased protein carbonylation in cell lysate at 24 h (**C**). Memcode reversible stain for 15 µg protein depicted for 8 h (**B**) and 24 h (**D**) exposure. Statistical difference from 0 mg I/mL diatrizoic acid loading for gels is depicted by an asterisk (* *p* < 0.05, ** *p* < 0.01, *** *p* < 0.001). Values represent mean ± SEM for three independent experiments.

**Figure 12 ijms-20-04074-f012:**
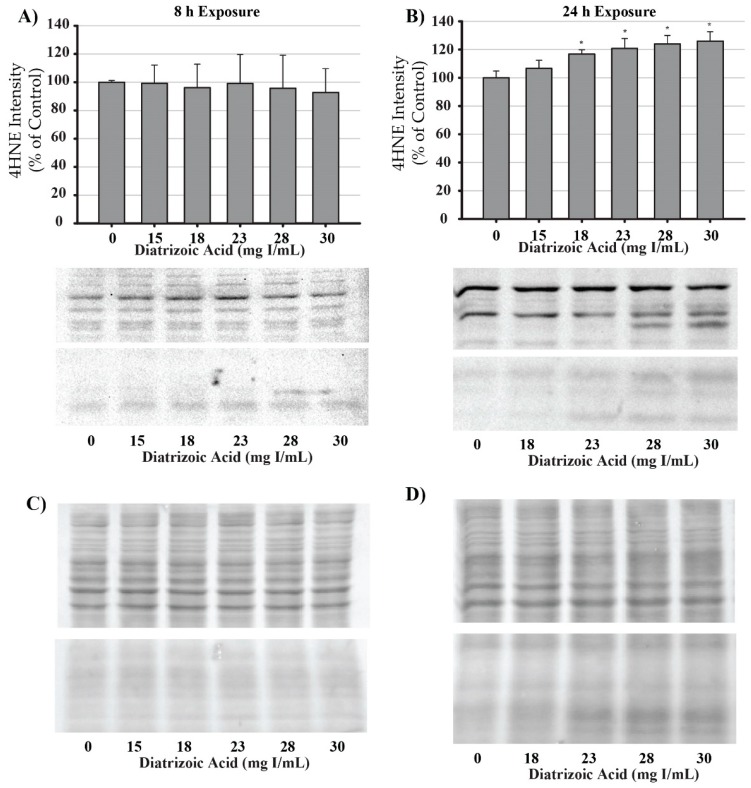
Diatrizoic acid effects on 4-hydroxynonenol (4HNE) adduct formation in HK-2 cells. An increase in 4HNE protein adduct formation was evident in cell lysate following 24 h exposure (**B**) to 18–30 mg I/mL. Positive bands of 4HNE were unchanged after 8 h (**A**) DA exposure. Panels (**C**) and (**D**) depict 8 h and 24 h, respectively, for protein loading of 40 µg per lane. Statistical difference from 0 mg I/mL diatrizoic acid depicted by an asterisk (* *p* < 0.05). Values represent mean ± SEM for three independent experiments.

**Figure 13 ijms-20-04074-f013:**
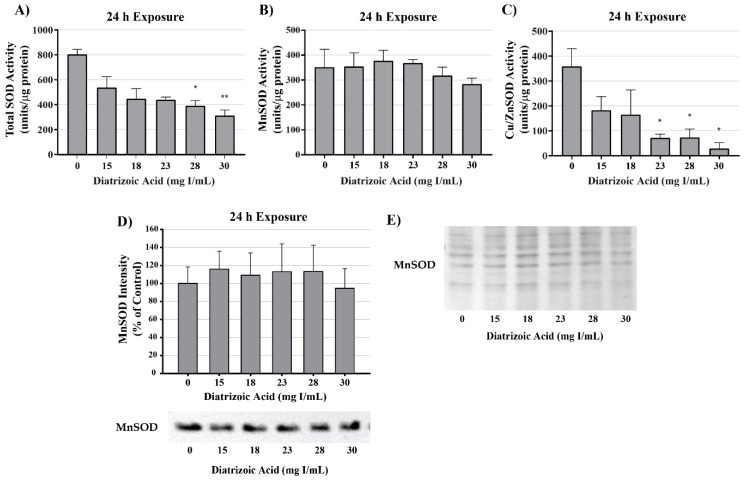
Diatrizoic acid effects on superoxide dismutase expression and activity in cellular fractions of HK-2 cells. Total superoxide dismutase (SOD) activity was decreased at 24 h by 28 and 30 mg I/mL (**A**). DA exposure did not change MnSOD activity (**B**) or protein expression (**D**). A decrease in Cu/Zn activity (**C**) was evident at 24 h exposure to 23–30 mg I/mL. Protein staining with Memcode (**E**) shown for blot visualizing MnSOD (D). Statistical difference from 0 mg I/mL diatrizoic acid depicted by an asterisk (* *p* < 0.05, ** *p* < 0.01). Values represent mean ± SEM for three independent experiments.

**Figure 14 ijms-20-04074-f014:**
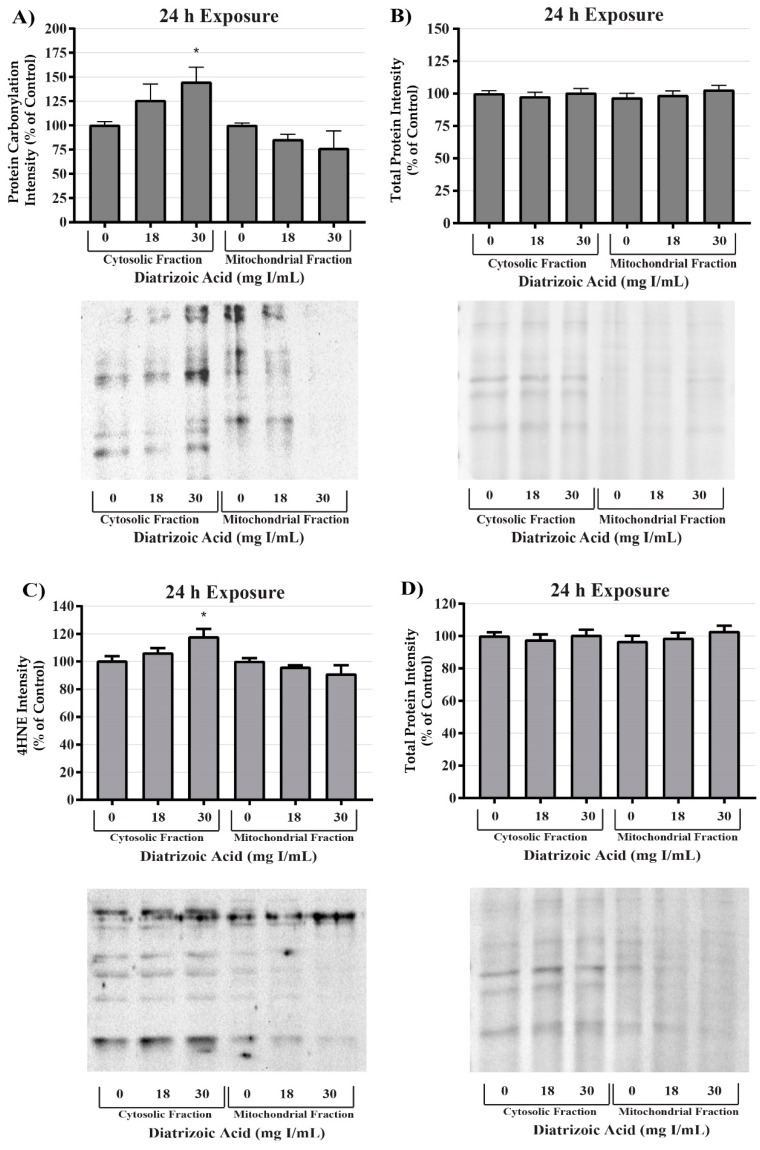
Diatrizoic acid effects on oxidative stress within cellular fractions of HK-2 cells. DA-induced oxidative stress predominantly in the cytosol. Representative blot and cumulative densitometry for expression of Oxyblot (**A**) and 4HNE protein adducts (**C**) in cytosolic and mitochondrial fractions following 24 h exposure to DA. Memcode reversible staining of blots loaded with 15 and 40 µg protein and densitometry for Oxyblot (**B**) and 4HNE (**D**). Asterisks (* *p* < 0.05) indicate statistical difference from vehicle control in cytosolic fraction. Values represent mean ± SEM for three independent experiments.

**Figure 15 ijms-20-04074-f015:**
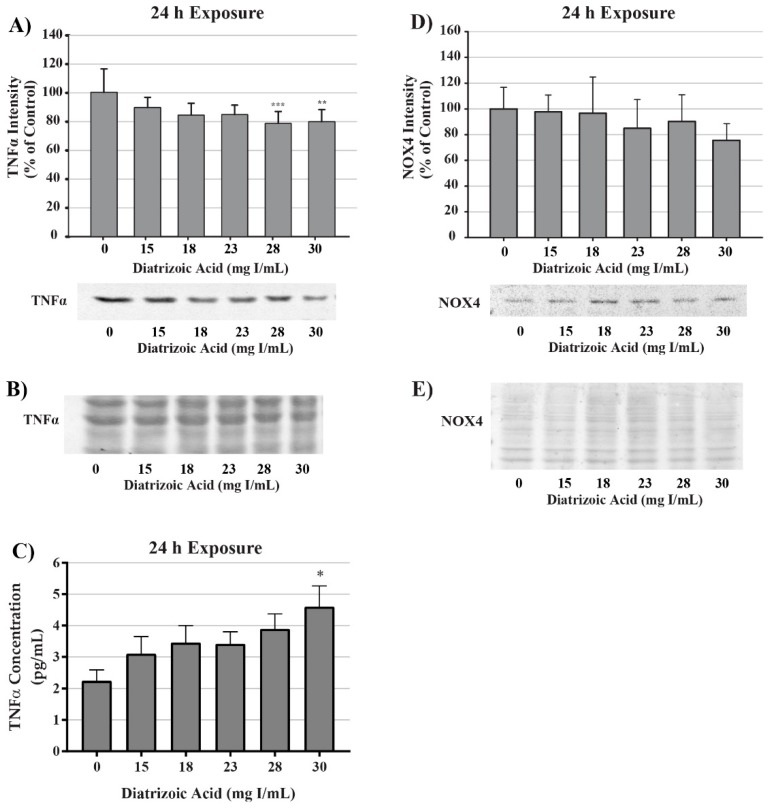
Diatrizoic acid effects on tumor necrosis factor alpha (TNFα) and NADPH oxidase (NOX4) expression in HK-2 cells. DA induced TNFα activation but did not affect downstream effectors. TNFα expression in cell lysate was decreased in response to 28 and 30 mg I/mL DA at 24 h (**A**). An increase in TNFα leakage into the cell media was evident at 24 h (**C**) with 30 mg I/mL DA. No significant change in NOX4 expression was apparent in cell lysate following 24 h (**D**) exposure to DA. Representative blot with Memcode reversible stain for 40 µg loaded protein depicted for TNFα (**B**) and NOX4 (**E**) following 24 h exposure. Statistical difference from 0 mg I/mL diatrizoic acid depicted by an asterisk (* *p* < 0.05, ** *p* < 0.01, *** *p* < 0.001). Values represent mean ± SEM for three independent experiments.

**Figure 16 ijms-20-04074-f016:**
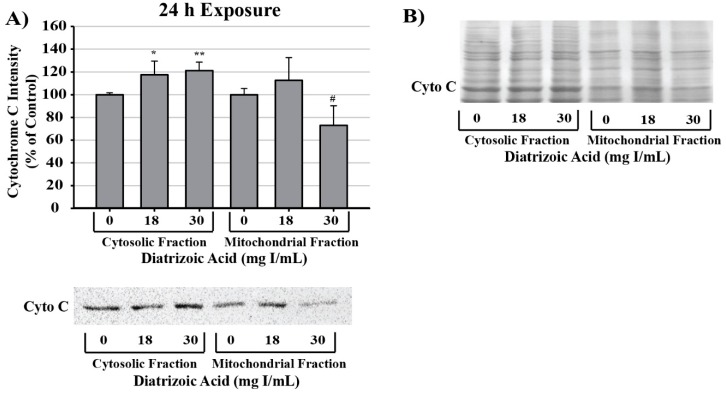
Diatrizoic acid effects on mitochondrial membrane integrity in HK-2 cells. DA diminished mitochondrial membrane integrity. Representative blot and cumulative densitometry for cytochrome c expression (**A**) in cytosolic and mitochondrial fractions following 24 h exposure to DA. Representative blot of protein staining to confirm equal 40 µg protein loading for cytochrome c (**B**). Asterisks (* *p* < 0.05, ** *p* < 0.01) indicate statistical difference from vehicle control in cytosolic fraction. Octothorpe (# *p* < 0.05) indicate statistical difference from vehicle control in mitochondrial fraction. Values represent mean ± SEM for three independent experiments.

**Figure 17 ijms-20-04074-f017:**
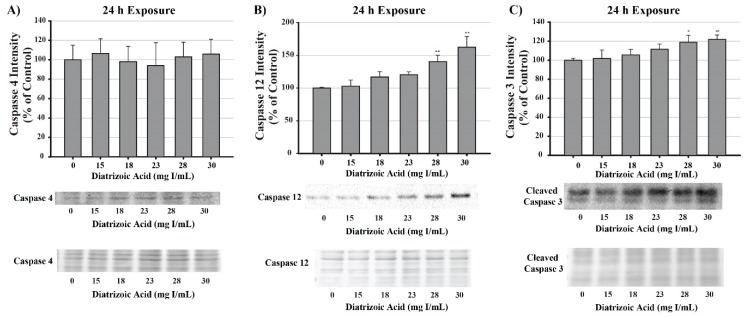
Diatrizoic acid effects on the expression of caspase 4, caspase 12, and caspase 3 in HK-2 cells. DA induced apoptosis via caspase 3 and caspase 12 activation. Representative blot and cumulative densitometry for total caspase 4 (**A**), caspase 12 (**B**), and cleaved caspase 3 (**C**) protein expression following 24 h exposure to DA. Asterisks (* *p* < 0.05, ** *p* < 0.01) indicate statistical difference from vehicle control. Protein loading reversible stain shown below respective protein. Each lane was loaded with 40 µg protein. Values represent mean ± SEM for three independent experiments.

**Figure 18 ijms-20-04074-f018:**
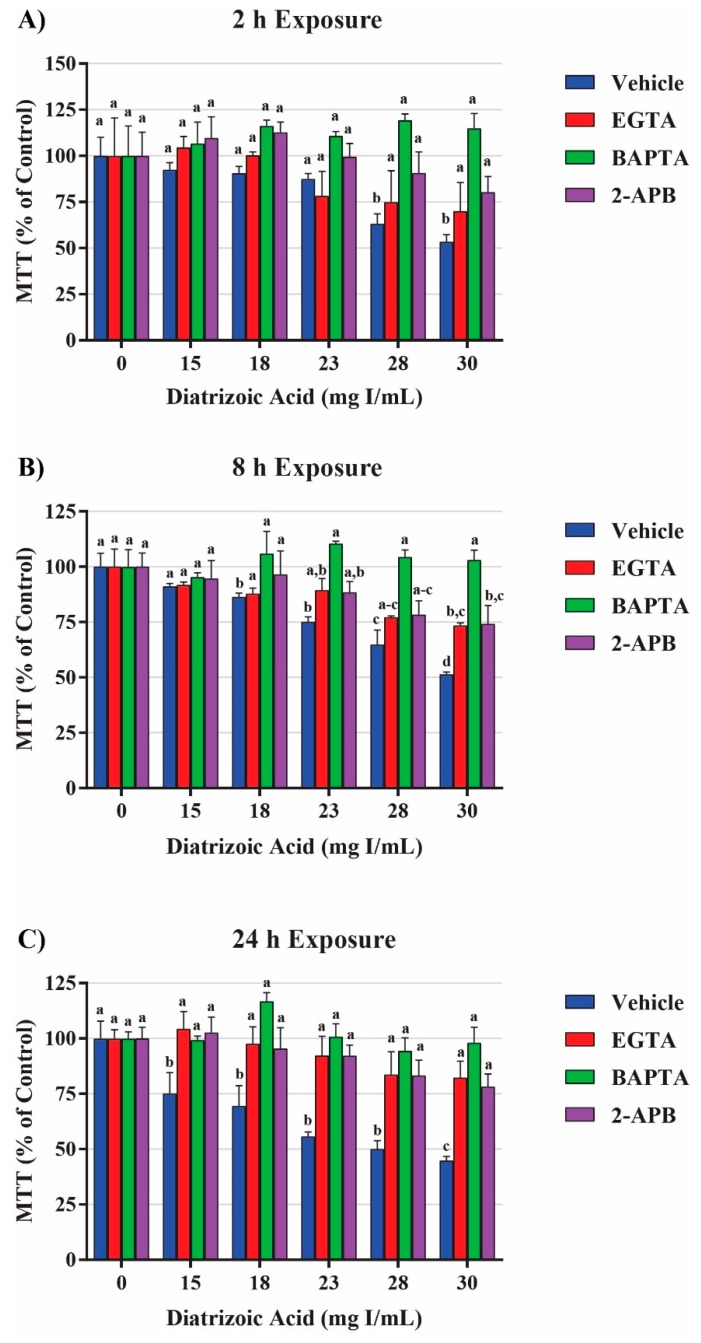
Effects of EGTA (extracellular calcium chelator), BAPTA-AM (intracellular calcium chelator), or 2-APB (inositol triphosphate receptor antagonist) pretreatment on mitochondrial viability in HK-2 cells. DA induced cytotoxicity was attenuated in response to calcium concentration modulators. BAPTA-AM offered protection from DA-induced cytotoxicity within 2 h (**A**), 8 h (**B**), and 24 h (**C**). EGTA and 2-APB provided partial protection from DA-induced cytotoxicity within 8 h (**B**) and 24 h (**C**). Different superscripts (a–d) indicate a statistical difference (*p* < 0.05) between groups. Values represent mean ± SEM for three independent experiments.

**Figure 19 ijms-20-04074-f019:**
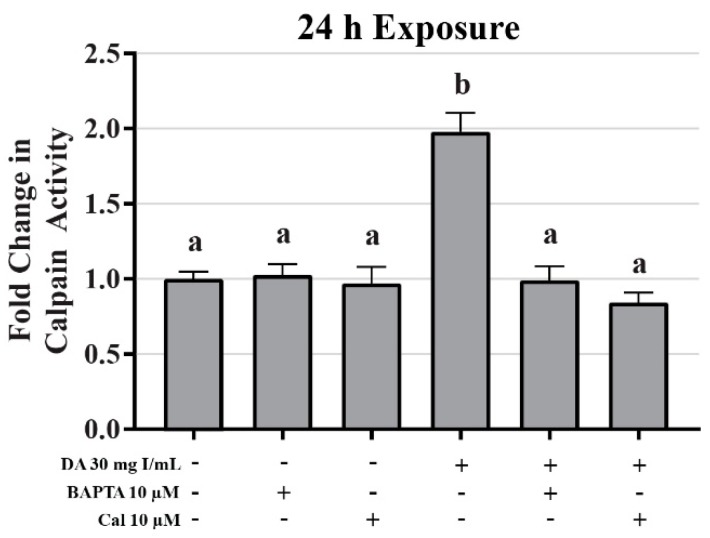
Diatrizoic acid effects on calpain activity in HK-2 cells. DA induced activation of calpain. HK-2 cells exposed to 30 mg I/mL DA for 24 h demonstrated a two-fold increase in calpain activity. Pretreatment with either 10 µM BAPTA-AM (calcium chelator) or 10 µM calpeptin (Cal; calpain pathway inhibitor) completely abrogated DA-induced calpain activity. Different superscripts (a,b) indicate a statistical difference between groups. Values represent mean ± SEM for three independent experiments of two biological replicants.

**Figure 20 ijms-20-04074-f020:**
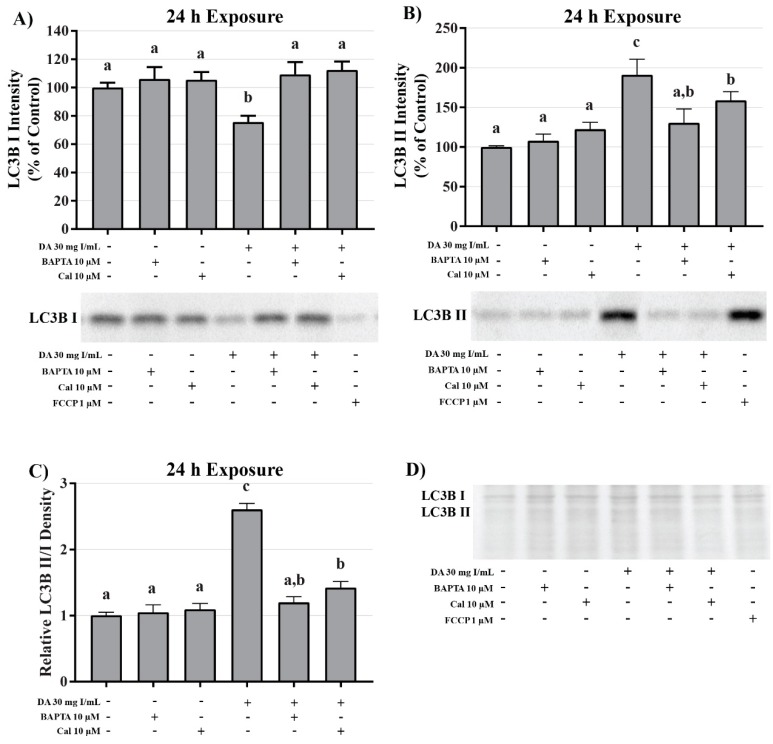
Effects of BAPTA-AM or calpeptin pretreatment on light chain (LC)3B expression in HK-2 cells. Pretreating HK-2 cells with BAPTA-AM or calpeptin attenuated DA-induced conversion of LC3BI to LC3BII. Representative blots and cumulative densitometry included for LC3BI (**A**), LC3BII (**B**), and LC3BII/LC3B I ratio (**C**) following 24 h exposure to DA. Protein loading stain depicted for 40 µg loaded pin (**D**). Positive control for LC3B conversion was FCCP (oxidative phosphorylation uncoupling agent). Different superscripts (a–c) indicate a statistical difference (*p* < 0.05) between group. Values represent mean ± SEM for three independent experiments of two biological replicants.

**Figure 21 ijms-20-04074-f021:**
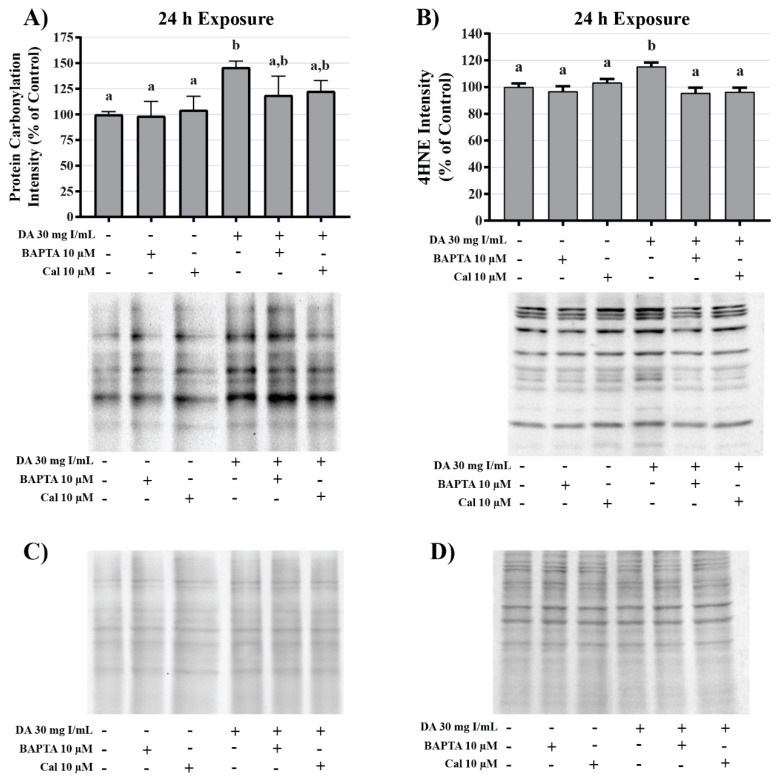
Effects of BAPTA-AM or calpeptin pretreatment on oxidative stress in HK-2 cells. Pretreatment with BAPTA-AM or calpeptin slightly decreased protein carbonylation and completely abrogated 4HNE adduct formation. Representative blots and cumulative densitometry included for protein carbonylation (**A**) and 4HNE adduct formation (**B**) following 24 h exposure to DA. Panel (**C**) depicts protein loading of protein carbonylation membrane and panel (**D**) depicts protein loading for 4HNE membrane. Different superscripts (a,b) indicate a statistical difference (*p* < 0.05) between groups. Values represent mean ± SEM for three independent experiments.

**Figure 22 ijms-20-04074-f022:**
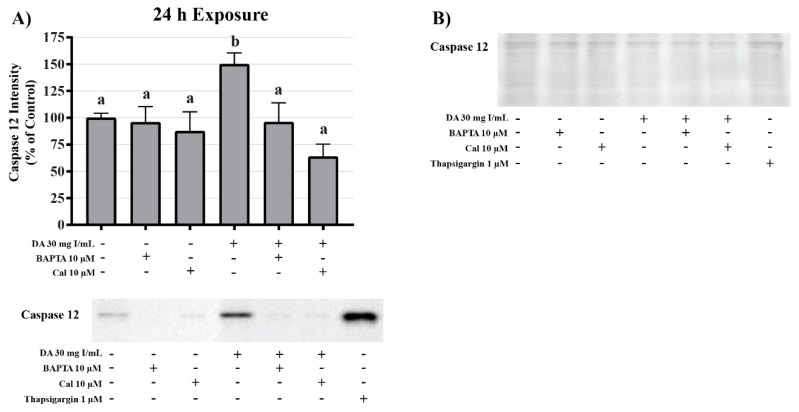
Effects of BAPTA-AM or calpeptin pretreatment on caspase 12 activity in HK-2 cells. Pretreatment with BAPTA-AM or calpeptin abrogated caspase 12 activation. Representative blots and cumulative densitometry included for caspase 12 (**A**) following 24 h exposure to DA. Representative blot with Memcode reversible stain for 40 µg loaded protein (**B**). Positive control for caspase 12 activation was thapsigargin. Different superscripts (a,b) indicate a statistical difference (*p* < 0.05) between various groups. Values represent mean ± SEM for three independent experiments.

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
