# Peer review of "Radiocontrast Agent Diatrizoic Acid Induces Mitophagy and Oxidative Stress via Calcium Dysregulation"

_ijms, 2019, doi:10.3390/ijms20174074_

Round 1

Reviewer 1 Report

Ward et al. present a detailed analysis of the effects of the diatrizoic acid on HK-2 cell metabolism. They choose concentrations of DA also obtained in vivo and use HK-2 cell line as model system for human proximal tubular cells. Therefore, their data have translational relevance.

A broad spectrum of parameters is determined, which may lead to confusion if not presented in an optimal manner. This is outlined in detail below. The discussion needs to be shortened and more focused. Subheadings might be helpful.

Presentation of data:

Figs. 1, 2, 18, 20, 21, 22: Superscripts to indicate statistical differences are not explained.

Western blots should be evaluated in standard manners: For each blot the ratio of the densitometric values of the specific bands and the respective control bands (in this case: quantification of Meme code reversible staining) should be calculated. The ratio of the control band (non-treated cells) should then be set to 1 or 100 % and used to summarize data of several blots. This type of evaluation would also considerably reduce the number of graphs presented and contribute to the clarity of the presentation.

As many different parameters are presented in the figures it would be helpful to label the figures not only as ‘intensity’ but to add information on which intensity is presented.

Line 113: why was the increase in OCR linked to proton leak?

Results lines 138-148 refer to Figs 7 and 10, but seem to describe Fig. 5?

Fig. 5 is labeled % of control. However, the figure does not show 100 %. Therefore, the reference needs to be explained or the data need to be reevaluated.

Fig. 5d vs 5a: The example of mitochondrial ATP production shown in Fig. 5d does not seem to match the evaluation shown in Fig. 5a. In order to obtain the graph shown in Fig. 5a, the other two experiments must look very different (5d: 15 and 23 mg/mL show equal values).

Fig. 6c: Statistics should be provided as there seems to be a strong decrease at 28 mg/mL compared to control. Why was mitochondrial fuel oxidation measured only after 8 h and not after 24 h?

Fig. 9d: Comparing lane 3 and 4, the graph does not reflect the Meme code staining shown below.

Fig. 10a is labeled % of control. However, the figure does not show 100 %. How were the error bars calculated?

Fig. 11, Fig. 13d, Fig. 14 a,c, Fig. 15, Fig. 17: How was the error bar of the control intensity calculated?

Fig. 11: The authors might want to summarize the findings presented in lines 225-232 in a final sentence.

Fig. 12: Why are only parts of the blots shown? How were the areas selected?

Figs. 14 b/d: Meme code stain shows far less protein in the mitochondrial fractions compared to the cytosolic fractions. This is not reflected in the graphs, nor are both controls (cytosolic/mito) set to 100 %. This is only a minor point, because it will no longer be relevant once the blots are evaluated lane by lane as outlined above.  

Fig. 15 c: There seems to be a high variability in TNFalpha release from the cells. Perhaps it would be more conclusive to normalize the data to control cells to excluded variations due to cell culture variability? Otherwise, a time course might help to show the increase in TNFalpha in a more convincing manner.

Fig. 16. The blot shown does not support a significant increase in cytochrome c in the cytosolic fraction. In order to support these data, the authors should show a time course of cytochrome c release.

Fig. 17 shows expression of different caspases. The graphs should be labeled accordingly. In Fig 17c, presentation of the uncleaved form of Casp3 is missing.

Fig. 21: BAPTA/AM seems to increase protein carbonylation rather than do decrease it? As long as the superscripts are not explained it is difficult understand the description ‘slightly, not to a significant degree’. The part of the blot evaluated in Fig. 11 seems to differ from the part shown in Fig. 21. Please show molecular weight markers.

Discussion and general points:

The authors make a strong statement that they used clinically relevant concentrations of DA. Therefore, they need to make clear what they consider clinically relevant. As outlined in lines 421 – 432, there is a considerable variability in the clinically obtained values. Numbers are given, but the reader is expected to calculate the plasma concentration of 300 mL of a 76% DA injection? This is not acceptable. It should be explained, why 40 mg/mL are considered to be toxic, whereas 30 mg/mL are considered to be non-toxic (lines 598 and following). In order to explain the discrepancies between their data and the data published by Peng et al., the authors need to perform experiments with 40 mg/mL DA. Even though the same cell line was used in both studies, reactivity of HK-2 cells may vary largely depending on the culture conditions. Therefore, it order to line up with the literature the authors need to clarify whether there is indeed such a strict concentration dependence in HK-2 cells in vitro.

The discussion should be shortened considerably and be more focused on the results obtained. E.g., the detailed description of the divergent roles of intracellular calcium is interesting, but does not seem necessary for the understanding of the manuscript (lines 611-695). Otherwise, more detailed studies related to the source of calcium (line 692/693) or other mechanistic aspects of calcium metabolism should be performed.

In some of their citations the authors refer to other cells than epithelial cells. As metabolism may vary considerably between different cell types, and as HK-2 cells are a model system of proximal tubular epithelial cells, which are very specialized epithelial cells, the authors might want to restrict the discussion to epithelial cells.

In line 454, the authors state that ‘HK-2 cells have similar activity to in situ PT cells’. Which data do they refer to? Even though HK-2 cells have many characteristics in common with primary cells, they are still a cell line with all the advantages and disadvantages of immortalized cells.

Conclusion

The conclusions are largely repetitive of the results obtained. Rather than referring to individual methods the authors should summarize the take home message, differentiating between evident results and open questions.

Author Response

The authors wish to thank the reviewer for their thorough evaluation of our manuscript. The Discussion has  been shortened based on the comments of the reviewer. The figures and figure legends have been revised to improve clarity of our findings.

Figs. 1, 2, 18, 20, 21, 22 Figure legends were modified to include the statistical p value.

The Western blot analysis was done via standard methods as described by the reviewer.

The figure Y-axes have been modified for the western blots to improve clarity of analysis. For example Figure 7 and LC3BI is Panel A and LC3BII is panel B. Figure 9 y-axis if GRP78 (% of Control).

Line 114 addresses the increase in proton link suggests mitochondrial dysfunction. The OCR difference between total complex inhibition and OCR following addition of oligomycin is proton leak. The increase in OCR linked with proton leak was evident at 24 h with the 30 mg I/ml DA.

Line 139 modified to Figures 1 and 2

Figure 5 has been modified according to the reviewer’s comments. Fig5D is a representative graph of one ATP-rate assay. Fig5A indicates that the change in mito-ATP production between vehicle and our 30 mgI/ml concentration was approximately 15 pmol O2/min. The difference between 15 mg I/mL and 28 mg I/mL in Fig5D is about 5 pmol O2/min which is similar from the statistics. The other experimental graphs look similar.

Figure 6 The statistics used for every experiment was a one-way ANOVA followed by a holm-sidak post-hoc test (aside form the MTT and Trypan blue data). The power of this statistical analysis did not indicate significance. Mito fuel flex tests were not performed at 24h because mitochondrial damage was occurring at 8h in the absence of fuel oxidation. Running this test at 24h would not indicate the source of toxicity.

Figure 9 The memcode shown was matched to the blot shown.

Figure 10 y-axis was corrected.

Figures 11-17 the error bars were SEM. All data in manuscript is expressed as SEM.

Lines 239-241 A summary sentence has been added regarding oxidative stress.

Figure 12 The 4HNE at 8 and 24 h was depicted at the 17 and 72 kDa bands as these were the only areas with changes between groups.

Figure 14 Equal protein loading for all lanes was done for cytosol and mitochondria and the densitometry showed no differences between groups for the Memcode. The profile of proteins would change depending on the protein location between the cytosol and mitochondria.

Figure 15 The release of TNFa into the media was measured by ELISA. Although there was variability, we did not observe  there were no downstream changes in NOX4 detected and we chose to not pursue a time study for  TNFa.

Fiugre 16 depicts a decline in cytochrome c expression in mitochondrial fractions and an increase in the cytosol and this occurred only at 24 h. Examining earlier times would only show no change.

Figure 17 has been revised. There was no change in total caspase 3 so it was omitted.  

Figure 21 BAPTA-AM does not induce protein carbonylation and was statistically not different  BAPTA-AM partially protects HK-2 cells from the DA-induced increase in protein carbonylation.

MW added to figure

Line 445 Our apologies. The manuscript has been modified “300mL of a 76% DA preparation or equivalent has been injected.. approximately 30 mgI/ml.

The Discussion has been significantly shortened.

The other cell types were added due to the limited number of studies.

Line 457 references 15, 17 have been added for in situPT cells.

The conclusion has been modified.

Reviewer 2 Report

This is an interesting and well-written manuscript describing a series of experiments designed to assume mechanisms of radiocontrast-induced nephrotoxicity with focus on mitochondrial functioning. There is no doubt that authors are very competent in the field of nephrotoxicity and the amount of work done is very impressive. Authors has some interesting findings with the effects of Diatrizoic Acid on cells bioenergetics. However, the paper has numerous serious concerns that limit this reviewer’s enthusiasm.

First of all, data presentation feels rather “raw” and are very hard to analyze and interpret. Below are lists of suggestions for article structure as well with some questions.

Major comments:

1)    Figure 2. Given concentrations of Diatrizoic Acid (DA) have no effect on cell viability (except high concentrations and long exposure). However data from Seahorse experiments and MTT suggest that mitochondria functioning was impaired even by low doses on 8 hours exposure. It is interesting to see if there will be a difference in cell viability after 2/8 hour exposure to 15 mg/ml DA but measured on 24 hours or/and 48 hours. It is possible, that on 24 (or even 48) hours there would be a visible decline in cell viability even if exposure to DA itself would be as short as 8 h.

2)    Figure 3. There is some contradiction in data of Fig 3D and Fig 3B. It seems from panel 3D that 5 mg/ml (green line) has a bigger effect, that 15 mg/ml (red). It is strange by itself, but it also does not align with panel B data. For example, it seems that basal respiration of 5 mg/ml is lower than 15 mg/ml, which should be almost identical to control. Which is not the case. Please clarify.

3)    Why in figure 3A 5 mg/ml concentration is missing? Especially since it behaves so strangely on 24h panels.

4)    LC3 (isoforms) levels is discussed in the context of mitophagy. However, it is conventional marker of autophagy and even in this role is not considered to be 100% reliable. It is indeed a good “pointer” to possible mitophagy, but it needs more investigation and proves, such as mitochondria number in cell, levels of PINK/parkin, or mitochondrial protein (cytochrome oxidase, VDAC, etc).

5)    SOD activity is discussed in the context of antioxidant capacity. Experiments addressing total antioxidant capacity (Trolox equivalent antioxidant capacity) would be interesting in this context.

6)    On some blots “representative” images seems to be not representative. For example on figures 7A, 7B, 9C, 15A or 17C histograms do not correspond to images. Differences in bands dencity looks stronger in blots than in diagrams. Whether it is possible to find more presentative images? Could you provide raw blots data, may be in supplements.

7)    Figure 18. Data is presented already normalized. It would be nice to see absolute numbers (at least for control) for comparison. For example, if we will use some toxic compound which will almost kill cells, its normalized control 0 mg/ml DA 100% (which is actually “all dead cells” from biological point of view) value won’t change at all no matter what we do with cells (since they are already dead).

Minor comments:

1)    On many images, text (axis labels, concentrations, scale) is barely readable. This certainly needs addressing.

2)    It would be helpful to add on the figures some text describing the content (and not only in figure description). Since most of the data is presented in the form of histograms (which have even the same groups and times) it is quite difficult to navigate through all 22 figures without some easily understandable label  like “4HNE level”, or “TNFa signal” and so on, right on the histogram axis or on the blots.

3)    In general, it seems that data presentation can be much more compact, without losing information. For example, all blot figures have cumulative protein signal. It is suggested to move these data to supplements, or delete.

4)    There are also quite a lot of negative data presented. Maybe authors can think of a way to present it in a more compact manner (tables; mention it only in the text?) while moving images for it to supplements.

5)    It feels that some figures (like 7 and 8, 11 and 12) are possible to merge, especially if some other suggestions on data compactization would be accepted.

6)    It seems that some blots (like fig7 a, b) are cropped from the same membrane. If it is true, it would be less confusing (as well as more compact) to see it on the same image.

7)    The discussion section is redundant. Authors should try to make it more concise.

8)    Presenting so much data authors should give some short conclusion and maybe graphical abstract, summarizing their findings.

Author Response

Figure 2 We conducted trypan blue exclusion at different time points. Only the 24 h showed any differences following DA treatment.

Figure 3D green line is 15 mg I/ml and the red is 5 mg I/ml.

Figure 7 has been revised. Further exploration of mitophagy is planned but our goal for this manuscript was to explore potential events associated with DA exposure.

Figures for western blots have been revised to reflect changes as percent of vehicle control.

Figure 9 the Memcode was matched to the blot depicted.

Figure 10 y-axis has been corrected.

Figure 18. The purpose of these experiments were to evaluate whether extracellular calcium chelation by EGTA or intracellular calcium chelation with BAPTA-AM could impact DA cytotoxicity. The net changes for vehicle, DA alone, chelator alone and the combination of DA+chelator provides the best comparisons. There were no differences between groups for MTT and pretreatments so the chelators EGTA and BAPTA-AM at the concentrations tested in this manuscript, were similar to vehicle.

The discussion has been condensed in the revision.

Reviewer 3 Report

All the blots must be improved , and presented again.

And more detailed comments:

1- Some abbreviations are not fully explained
2-lines 44-47 ; references need to be specified for each damage
3-would the significance of the results will be lost if the SD deviation was used instead of the SEM?
4-the early apoptosis withing the first 8 hr need to be explained or discussed.
5- some blots are highly croped or no specific to be quantified, it need to be improved
6-Cells can be directly stained with DCF or DHE to examine the ROS production.
7-point of NOX4 expression need to be more explained , the higher expression is not the only marker for the higher activation.
8-NOX 2 is also expressed in the HK-2 , why the author did not discuss it?

Author Response

Reviααewer #3αα

1-The authors would be glad to provide meanings for missing abbreviations but the reviewer did not provide a list of undefined abbreviations.

2- Lines 44-47 The list of reference [8-12] provide the characterization of contrast nephrotoxicity. References [8-9] describe vacuolization and this has been added to the revised manuscript.

3-The statistical analysis and conclusions would not change in the results if the data was expressed as SD instead of SEM. We have kept the results expression as SEM.

4-Apoptosis did nor occur at 8 h. We found a decline in MTT at 2, 8 and 24 h (Figure 1) but cleaved  caspase 3 was not detected until  24 h (Figure 17).

5-We have improved the blots to provide the specific MW and protein of interest.

6-The authors are aware that examining ROS with DCF would provide additional characterization of DA changes to HK-2 cells. However, our findings suggest that oxidative stress and modifications of proteins occurs at 24 h while changes in the mitochondria and MTT occur within 2 h. Consequently we did not measure ROS.

7-The authors measured NOX4 as NOX4 is a major contributor of oxidative stress by   TNFα.

8-NADPH oxidase 4 (NOX4) is the predominant NOX expressed in human proximal tubules (Sedeek et al., 2013;  JASN 24 (10) 1512-1518). Consequently, we focused on NOX4. Future studies perhaps may examine expression of  NOX2 but this was beyond the scope of our studies.

Round 2

Reviewer 1 Report

Figs. 1, 2, 18, 20, 21, 22 - lettering is not explained. What means a or b or c etc?

The authors claim that they reevaluated the Western blots. How come the graphs did not change? They still do not show the ratio protein of interest to house keeping protein, but protein of interest and total protein in separate graphs.

The authors still do not explain the meaning of the error bar of control lanes in Western blots - how was the error bar calculated. It is not sound to use absolute densitometric values due to technical variability.

In figure 10, control values were graphically shifted to 100 %. Usually, one can not simply change graphs.

In several figures the labeling of the axes waschanged - obviously, there were quite some mistakes in version 1.

Fig. 15 Error bars of TNFa are smaller in the new version - why??

The difference between 30 and 40 mg/ml - clinically relevant vs toxic it not discussed.

The conclusion has been modified: Conclusion in version 2 is identical as conclusion in version 1.

Author Response

Figs. 1, 2, 18, 20, 21, 22 - lettering is not explained. What means a or b or c etc?

RESPONSE: We have designated what the different letters indicate in the figure legends. The figure legends state that different superscripts (figure 1 a-f) indicate a statistical difference between groups across all time points. Therefore, treatment with superscripts of “a” are not statistically different from each other while treatments with superscripts of “a” and “b” are statistically different from each other at p<0.05.This designation has been used in other manuscripts published in IJMS see Figures 1 and 2 in Lee et al., IJMS 2019, 20, 2443; doi:10.3390/ijms20102443.

The authors claim that they reevaluated the Western blots. How come the graphs did not change? They still do not show the ratio protein of interest to house keeping protein, but protein of interest and total protein in separate graphs.

RESPONSE: Housekeeping proteins were not used for normalization. This is due to the fact that common housekeeping genes such as actin and GAPDH can be up- or downregulated in response to xenobiotics and differing physiological states. As a result, total protein is a better “house keeper”. Please see a) Dwinovan et al., Proteomic analysis reveals downregulation of housekeeping proteins in the diabetic vascular proteome Acta Diabetol 54:171-90. 2017; b) Arukwe A. Toxicological housekeeping genes: do they really keep the house? Environ. Sci. Technol. 40: 7944-7949, 2006 and c) Hu et al., Common housekeeping proteins are upregulated in colorectal adenocarcinoma and hepatocellular carcinoma, making the total protein a better "housekeeper. Oncotarget 7(41): 66679-66688, 2016.

The authors still do not explain the meaning of the error bar of control lanes in Western blots - how was the error bar calculated. It is not sound to use absolute densitometric values due to technical variability.

RESPONSE: The control value is calculated for each gel normalized to protein. The average for all controls from all gels is set as the average and assigned 100%. The variability between controls provided the SEM for the control bar. This has been a method used in other per reviewed publications (see Seguella et al., IJMS 20(13), 3240, 2019 https://doi.org/10.3390/ijms20133240 Figure 1; Chen et al., Biomed and Pharmacother 106:1175-1181, 2018 Figure 2).

In figure 10, control values were graphically shifted to 100 %. Usually, one can not simply change graphs.

RESPONSE: Following a rigorous re-evaluation of the statistics, it was noticed that a mistake was made and not all the values for control (0 mg I/mL DA) were used in generation of the graph. Once the missing values were added, the control graph shifted back to 100%.

In several figures the labeling of the axes was changed - obviously, there were quite some mistakes in version 1.

RESPONSE: There were some errors in version 1 which are now corrected.

Fig. 15 Error bars of TNFa are smaller in the new version - why??

RESPONSE: Upon further inspection of the statistical analysis, it was noticed that the error bars in the original transcript reflected SD instead of SEM. The error bars were changed to SEM to remain consistent with the presentation of the rest of the data. Below is a table reflecting the Mean, SD and SEM. The authors apologize for missing this inconsistency.

Reviewer 2 Report

Some of the data in the figures are redandant. It woulb be better to keep only the results indicating significant changes of the target protein(s) on Western blot. Whereas the data of loading protein controls (the images of Memcode Reversible staining and/or diagrams of their densitometry indicating that protein load is the same) could be removed from the figures.

Author Response

Thank you for your review. We are hesitant to remove the protein loading images in light of reviewer #1's comments. Any thoughts?

Reviewer 3 Report

The authors presented enough effort to answer the comments

Author Response

Thanks you for your review of our revised manuscript.